

# Characterization of fractures in potential reservoir rocks for geothermal applications in the Rhine-Ruhr metropolitan area (Germany)

Martin Balcewicz[1,2], Benedikt Ahrens[3], Kevin Lippert[3,2], and Erik H. Saenger[1,3,2]

[1]Department of Civil and Environmental Engineering, Bochum University of Applied Sciences, Lennershofstraße 140, 44801 Bochum, Germany
[2]Institute of Geology, Mineralogy, and Geophysics, Ruhr-University Bochum, Universitätsstrasse 150, 44801 Bochum, Germany
[3]Fraunhofer IEG - Institution for Energy Infrastructures and Geothermal Energy, Lennershofstraße 140, 44801 Bochum, Germany

**Correspondence:** Martin Balcewicz (martin.balcewicz@hs-bochum.de)

**Abstract.** The importance of research into clean and renewable energy solutions has increased over the last decade. Geothermal energy provision is proven to meet both conditions. Therefore, conceptual models for deep geothermal applications were developed for different field sites regarding different local conditions. In Bavaria, Germany, geothermal applications were successfully carried out in carbonate horizons at depth of 4000 to $6000\,\mathrm{m}$. High permeability rates combined with sufficient

thermal conductivities were mainly studied in karstified carbonates from the Late Jurassic reef facies. Similar to Bavaria, carbonates are located in the east of the Rhenohercynian Massif, in North Rhine-Westphalia (NRW), which quantification of the geothermal potential is still lacking. Compared to Bavaria, a supraregional carbonate mountain belt is exposed at the Remscheid-Altena anitcline (NRW) from late Devonian and early Carboniferous times. The aim of our study was to examine the potential geothermal reservoir by field and laboratory investigations. Therefore, three representative outcrops in Wuppertal,

Hagen Hohenlimburg, and Hönnetal were studied. During field surveys, 1068 discontinuities at various spacial scales were observed by scanline surveys. These discontinuities were characterized by trace length, true spacing, roughness, aperture, and filling materials. Joint orientation analysis indicated three dominant strike orientations in NNW−SSE, NW−SE, and NE−SW directions within the target horizon of interest. This compacted limestone layer (Massenkalk) is approximately $300\,\mathrm{m}$ thick and located at 4000 to $6000\,\mathrm{m}$ depth, dipping northwards at a shallow dip angle of about 30 to $40°$. An extrapolation of the measured

layer orientation and dip suggests that the carbonate reservoir extends below Essen, Bochum, and Dortmund. Our combined analysis of the field and laboratory results has shown that it could be a naturally fractured carbonate reservoir. We evaluated the potential fracture network in the reservoir and its orientation with respect to the prevailing maximum horizontal stress before concluding with implications for fluid flow: We proposed to focus on discontinuities that are approximately N−S oriented for upcoming geothermal applications, because the geothermal potential of the characterized reservoir matrix is insufficient

for deep geothermal applications. Our results indicate that even higher permeability can be expected for karstified formations related to the reef facies. Our compiled data set consisting of laboratory and field measurements may provide a good basis for 3D subsurface modeling and numerical prediction of fluid flow in the naturally fractured carbonate reservoir. Further studies





have to be elaborated to verify, if the fractured reservoir could possibly be reactivated by, for instance, hydraulic stimulation and thus enable geothermal applications.

# 1 Introduction

The Rhine-Ruhr metropolitan region in western Germany is one of the largest metropolitan areas in Europe. It has the largest European district heating network (AGFW, 2009) and is one of the largest energy systems in the world, which is predominantly fed by fossil fuels (Klaus et al., 2010). In recent decades, the Ruhr metropolitan area has been subject to structural changes associated with the ending of traditional coal mining and steel industry as well as accelerated urbanization. Considering effective climate changes, these processes pose local to global challenges for a post-fossil energy future. The polycentric structure of the Rhine-Ruhr area is particularly promising for the requirements imposed by the energy transition (Wegener et al., 2019). To account for the increasing demand for energy supply (e.g., Scheer et al., 2013; Araújo, 2014), the successful implementation of sustainable non-fossil energy concepts is required (e.g., geothermal energy; Fridleifsson et al., 2008; Goldstein et al., 2011). In recent years, many economically successful geothermal energy projects in Bavaria, southern Germany, have been realized and could serve as a valuable model for the Rhine-Ruhr area. The Bavarian part of the South German Molasse Basin has a high hydrothermal energy potential (Schellschmidt et al., 2010; Böhm et al., 2010; Stober, 2014; Homuth, 2014), which be can extracted from pre-existing fracture and karst systems in deep carbonate rocks (e.g., limestone deposits) of the Malm (Late Jurassic) formation (Fritzer et al., 2010). Similar to deep Bavarian deposits, carbonate rocks were accumulated in western Germany. Geothermal systems aiming for comparable geological carbonate horizons in the Rhine-Ruhr area are promising solutions to initiate the energy transition (Knutzen, 2017). The carbonates of the Rhine-Ruhr area were deposited extensively during Devonian times (Krebs, 1967; Koch, 1984; Koch-Früchtel and Früchtel, 1993). Today, those carbonates are exposed at the northern part of the Rhenohercynian Massif with thicknesses of approximately $300\,\mathrm{m}$ (Paeckelmann and Zimmermann, 1930; Paeckelmann, 1979; von Kamp and Ribbert, 2005). Based on limited exploration activities, these limestones are expected at depths relevant for deep geothermal applications (DEKORP Research Group, 1990). Younger, overburden sedimentary layers (Late Carboniferous) were deposited on top of the potential reservoir and the region became massively folded and faulted (Drozdzewski, 1985; Franke et al., 1990; Ziegler, 1990; Littke et al., 2008; Scheck-Wenderoth et al., 2008; Meschede, 2018). A feasibility study including extensive subsurface reservoir characterization and local geological site assessment is essential to describe the hydrothermal potential of presumably fractured and faulted carbonates in the Rhine-Ruhr area.

Determining the hydraulic, elastic, and thermal properties of a fractured carbonate reservoir is still associate with difficulties due to its accessibility, heterogeneity, and the in-situ conditions prevailing (i.e. stress and temperature). Even though very costly field measurements have been carried out, the characterization of the fractured reservoir is often complicated by insufficient field-based data, as they only provide a point by point insight into the reservoir properties (e.g., borehole logging; Da Prat, 1990). In addition, seismic surveys have been used for decades to indirectly estimate the properties and mechanical conditions of fractured reservoirs (e.g., Far, 2011); although, the interpretation for complex, heterogeneous reservoirs is not unique. In the best case all field methods can be combined to reduce the geological ambiguity to characterize the reservoir in the best



possible way. Since direct information on in-situ properties are limited and inaccessible (e.g., seismic tomography or well and borehole data), outcrop characterizations and laboratory measurements can provide the basis for reservoir characterization and the estimation of the potential reservoir behaviour under in-situ conditions.

The estimation and description of individual fractures and entire fracture networks in deep reservoirs is of elementary impor-
tance when it comes to reservoir characterisation. Fractures and fracture networks have a decisive influence on the fluid flow within the reservoir (e.g., Odling et al., 1999), but also on the stability (e.g., Cappa et al., 2005). They are typical observations at analogues outcrops and can be measured there (see review by Bonnet et al., 2001). While the origin of these typically tectonic driven fractures can be discussed extensively, the influence on reservoir permeability is obvious. Compared to the matrix permeability of the host rock, the permeability of fractures is generally several orders of magnitude higher (e.g., Nelson and
Handin, 1977; Kranz et al., 1979; Evans et al., 1997; Ahrens et al., 2018). Detailed fracture analysis is essential to model reservoir quality, optimal well locations, and well performance (Nelson, 2001). Field surveys are both the first and one of the most crucial steps in the exploitation of the reservoir (van Golf-Racht, 1982; Agosta et al., 2010), that is, the investigation of the discrete fracture network (DFN). Furthermore, establishing correlations between outcrop and laboratory observations provide the basis for an improved analysis of seismic surveys, borehole logs, and fluid injection experiments to be performed.

In this integrated study, we examine the geothermal potential of deep carbonates in the Rhine-Ruhr area, by combining field and laboratory investigations of pre-existing fractures and fracture networks on outcrop and sample scale. The combined analysis of field and laboratory measurements allows us to describe the potential carbonate reservoir by characterizing its natural fractured systems. Three outcrop analogues at the right-handed side of the northern Rhenohercynian Massif have been chosen for field survey and sample collection. All quaries are located within the Devonian Reef Complex and compacted
limestone (Massenkalk) from Middle and Upper Devonian are the dominant stratigraphic units in those quarries (Krebs, 1970; Paeckelmann, 1979; Schudack, 1993). Dominant fracture systems and different facies have been documented and sampled. Most of these measurements have been conducted by one-dimensional scanline surveys (Priest and Hudson, 1976, 1981). Representative fresh rock samples have been taken directly from the outcrop wall. Porosities, ultrasonic velocities, dynamic elastic moduli, thermal properties, and permeabilites were determined by laboratory measurements.

In the following, we first introduce the geology of the Devonian Reef Complex. We proceed by describing the outcrops and the employed field and laboratory methods. Following the presentation of the measurements results, we conclude by discussing the geothermal potential of the geological subsurface model, that is, the deep carbonate layers and their fracture systems.

## 2   Geology, Outcrop Investigations, and Laboratory Measurements

### 2.1   Geological Setting: The Devonian Reef Complex

The Rhenohercynian Massif is the European most northern mountain belt, which was defined after Kossmat (1927). The massif is the result of a shallow marginal sea which was surrounded by Laurussia in the North (Old-Red Continent and Baltica) and the micro-continent Avalonia in the South. During the Early Devonian, a passive continental margin evolved to the South of Laurussia which was affected by crustal thinning. On top of the thin crust a shelf sea derived called the Rhenohercynian



basin. The flat sea from Bretagne, Belgium up to Germany was filled by clastic shelf deposits and carbonates derived from

Laurussia. An overall tectonic NW-movement dominated during Hercynian Orogeny and lead to extensions, periodic trans-, and ingressions within the Rhenohercynian basin. In the late Middle Devonian, Avalonia moved northwards and the sedimentation in the offshore belt decreased. The ongoing NW-movement enhanced submarine volcanic mounds which favoured growth of carbonates and coral/stromatoporoid reefs on the shelf (Jux, 1960; Grabert, 1998; Dallmeyer et al., 2013; Franke et al., 2017). An elongated reef complex developed along Laurussia's coastline, the so-called the Devonian Reef Complex (Krebs, 1970).

The Devonian Reef Complex can be traced over an area from the Neandertal-Valley (Düsseldorf) to Dornap, Wuppertal, Schwelm, Hagen Hohenlimburg, Letmathe, Sundwig, Iserlohn, Hemer to Balve (Fig. 1). Furthermore, these horizons can be observed in boreholes in northern Germany (e.g., Hesemann, 1965, Münsterland 1). Continued crustal extension and subsidence are marked by a transition from siliciclastic, deltaic shallow-marine environments (Rhenish facies), to an open-marine shale and pure limestone dominated facies (Hercynian facies) (Krebs, 1970; Grabert, 1998; Pas et al., 2013; Meschede, 2018).

Shelf clastics and carbonates are overlain by hemipelagic deposits (Dallmeyer et al., 2013). During Late Devonian and Early Carboniferous, the reef complex receded northwards and sedimentation changed while closing the Rhenohercynian basin. Primary caused by tilting blocks and faulting, first graywacke turbidites and secondly a quartz-rich sandstone were deposited in the foreland. Further, during Late Carboniferous sedimentation continued and resulted in interbedded sequences of the Ruhr coal district at the northwestern margin of the Rhenohercynian foreland. From Early Carboniferous onwards, the deposited

sediments were compressed and folded by the Hercynian Orogeny. Simultaneously, the thick accumulations of sediments (3-12 km) were thrusted by reactivated listric normal faults during NW-movement (Oncken, 1997; Franke et al., 1990; Engel et al., 1983; Franke and Engel, 1982; Holder and Leversidge, 1986; Meschede, 2018). Between Early Triassic and Late Cretaceous at least two more extension sequences arose and caused NE-SW-directed shortening (Brix et al., 1988; Drozdzewski and Wrede, 1994). Since Eocene, the complex tectonic setting is affected by a further extensional regime (Kley, 2013).

**2.2 Study Areas**

Three stone pits have been chosen for field studies within the Devonian Reef Complex: (1) stone pit Osterholz located in Wuppertal, (2) quarry Oege in Hagen Hohenlimburg, and (3) quarry Asbeck in Hönnetal (Fig. 1). The outcrops in Osterholz, Oege, and Asbeck are active open pit mines and are driven by independent companies called Kalkwerke H. Oetelshofen GmbH & Co. KG, Hohenlimburger Kalkwerke GmbH, and Rheinkalk GmbH, respectively. The entrances of the three quarries are

located at (1) E $^{36}$3372, N $^{56}$79249, (2) E $^{40}$1215, N $^{56}$89459, and (3) E $^{40}$0334, N $^{56}$93287 (Fig. 2). In the following the outcrops will be referred according to their geographical location: Wuppertal (WOH), Hagen Hohenlimburg (HKW), and Hönnetal (HLO).

The quarry Osterholz in Wuppertal is located in the Osterholz graben which is limited by the Herzkamper syncline to the North and the Wupper river to the South (Fig. 1). The regional geology is dominated by WSW–ENE striking folds which

are associated with the Hercynian orogeny (Paeckelmann, 1979). The Herzkamper special syncline can be understood as an extension of the most-southern syncline in the foreland basin (Wittener syncline, Ruhr-Coal-District) but should be considered as an individual syncline separating the Remscheid-Altena anticline, in the South, from the Velbert anticline, in the North





(Paeckelmann, 1979). The compacted limestone deposits (Massenkalk) are located in an approx. 10 km long graben which strikes parallel to the Hercynian induced folds. This thrusted graben results into numerous smaller horses. The tectonic regime

of this system was mainly affected by the Hercynian orogeny and became further compressed by recent folding processes in the South. The stratigraphic unit of the quarry Osterholz in Wuppertal is Upper Givetian (approx. 382 Ma), primarily the Schwelm facies (according to Krebs, 1970). The carbonate layer (approx. 150 m thickness) sits on top of the dolomitic carbonates. These local limestone showed grayish, compacted layers of well-bedded carbonates with corals, stromatoporoidea, and bioclastic materials. Therefore, these limestone with a mean thickness of 1 to 5 m are either related to the brachiopoda- and coral rich

series or stromatopora series. Both series are indicators for the closer back-reef lagoon facies (Krebs, 1967; Koch, 1984; Grabert, 1998; Paeckelmann, 1979). However, dolomitic blocks also occurred in a small number.

The regional tectonic setting in Hagen Hohenlimburg was massively affected by the Remscheider anticline, the Ennepe-Thrust, and the Großholthausener fault (Fig. 1). In the South of Hagen Hohenlimburg several smaller scaled anticlines occur with lengths of 100 to 1000 m. These anticlines follow the Hercynian general striking direction of WSW−ENE and plunge steep to

the North with an angle of about 25°. Within these anticlines, smaller special folds are documented (von Kamp and Ribbert, 2005). In the northern limb of the Remscheider anticline, where the studied quarry was set, the Hasper anticline and Voerder syncline can be found. Both folds are monoclines and their steep limbs rapidly end when intersecting the regional normal fault Großholthausener fault. In the East of the Großholthausener fault, the quarry Oege in Hagen Hohenlimburg can be found. The quarry is surrounded by the river Lenne and some smaller normal faults. In the North, the devonian carbonate layers are cut

by the Ennepe-Thrust. Krebs (1970), Flügel and Hötzl (1976) and Koch-Früchtel and Früchtel (1993) characterized the local compacted limestone in the quarry Oege as Schwelm facies. The quarry is enriched in dolomite due to hydrothermal veins (von Kamp and Ribbert, 2005). These dolomites manifest in a high degree of fractured blocks and karst formations which were filled by tertiary- and cretaceous-sediments, respectively. The carbonates, which were untouched by the hydrothermal veins, contain brachiopoda, bivalves, crinoids, corals, gastropoda, stromatoporoidae, and micrite. These carbonates are identified as Middle

Devonian (Givetian according to Basse et al., 2016). The compacted limestone was mapped as irregular, lens shaped coral reefs (von Kamp and Ribbert, 2005). Those small reefs have been characterized as a preliminary stage of the today known Devonian Reef Complex. The complex distribution in the subsurface is still heavily debated (e.g., Salamon and Königshof, 2010). A detailed profile was taken by Koch-Früchtel and Früchtel (1993) and reveals compacted limestone with numerous dolomite layers (mean thickness 20 m). Below the Schwelm facies, layers of carbonatic siltstones and claystones, called the Oege Beds

(Upper Honsel Beds), are located. Furthermore, a full-microfacies analysis was made by Koch-Früchtel and Früchtel (1993) and subdivides the Schwelm facies into shallow to open sea lagoons which are affected by sea current.

The quarry Asbeck in Hönnetal is located to the North of the Remscheid-Altena anticline (Fig. 1). Due to its special location, the northern limb of the anticline, the geological regime can be described as a very basic setting. The exposed compacted limestone deposits are dipping gently to the north and are surrounded by older Devonian sediments in the South and early De-

vonian, late Carboniferous sediments to the North. The regional setting is dominated by NNW−SSE striking strike-slip faults which are associated to the folding mechanics during Hercynian orogeny (Paeckelmann, 1938). Those strike-slip faults can be tracked along the major Remscheid-Altena anticline structure. At the most eastern part of the Remscheid-Altena anticline, the





tip of the fold is plunging to the NNE direction. This affects the mentioned strike-slip faults which results into some bent fault
formations along the plunged tip of the anticline. During folding, weak bedding planes became thrusted and became reacti-
vated during later tectonic events. The local stratigraphy of the quarry Asbeck in Hönnetal consists of an initial bank stadium
(Schwelm Facies) that developed into a true bioherm complex, that is, Dorp Facies (Becker et al., 2016). Schudack (1993)
has investigated the different facies and cyclic depositions of the reef and back-reef lagoon facies and subdivided these facies
into six subfacies which were investigated in more detail by Kloke (2007). The grayish, highly compacted limestones (approx.
$600\,\mathrm{m}$ thickness) containing corals, stromatoporoidea and Brachiopoda are associated with late Middle Devonian (Givetian)
or early Late Devonian (Frasnian). Except some minor parts, that were mentioned by Kloke (2007), no dolomitic rocks were
documented.

The selected quarries show similar bedding planes with comparable dip directions. In the quarries Wuppertal, Hagen Hohem-
limburg and Hönnetal, mean bedding is approximately 327°, 345°, and 18° in dip direction with a dip angle of about 30 to
45°, 42°, and 28° in respect to dip direction, respectively. The working levels of each stone pit were approximately oriented
perpendicular to each other and strike NNW−SSE and NE−SW (Fig. 2).

## 2.3   Field Method: Outcrop Scanline Surveys

The term discontinuity and its origin can be looked up in many textbooks on structural and engineering geology (e.g., Priest,
1995). In the following, we define a discontinuity as a weak zone within a rock mass that can be referred to as a fracture, joint,
vein or fault. Unlike the International Society for Rock Mechanics (ISRM, 1978), we neglect weak bedding planes or schistosity
planes in the field surveys. Nevertheless, we distinguish between natural discontinuities, which are associated to geological
origins, and artificial discontinuities which result from anthropogenic processes (e.g., drilling, blasting or excavation).

The discontinuities were recorded by a scanline sampling technique according to Priest and Hudson (1981). The advantages
and disadvantages of this method have already been discussed in detail in several studies over the last decades (e.g., Attewell
and Farmer, 1976; Priest and Hudson, 1981). According to the study of Priest and Hudson (1981), this paper focuses only
on the most important parts of this technique. The setting of a scanline survey is a tape line, which is fixed firmly to the
exposed rock wall and shows a start and an end point. The length of a scanline can range from a few decimetres to several
metres. Estimates of all possible sources of error have shown that the number of measured and recorded discontinuities is more
significant than the length of the scanline (Priest and Hudson, 1981). Hence, for each quarry we selected representative, safe
rock walls where at least 30 measurable discontinuities (number of discontinuity observations $n \geq 30$) could be recorded. After
the tape line was fixed, the trend and plunge angle of the tape line were measured. The trend angle reflects the orientation of the
tape line ($\alpha_\mathrm{S} = 0° \equiv$ North; $\alpha_\mathrm{S} = 180° \equiv$ South), which corresponds to the strike of the rock wall. The angle of plunge shows
the levelling of the tape line (approx. $0° \leq \beta_\mathrm{S} \leq 15°$). After the tape line was installed, each discontinuity that intersected the
tape line was recorded and categorized according to the following six properties (cf. Table 1):

1. **Intersecting distance:** The intersecting distance corresponds to the distance between each intersecting discontinuity and
the fixed starting point of the tape line.





2. **Discontinuity length:** The installed tape line and the intersecting discontinuity meet at an intersecting point. The trace length above and below the tape towards the joint tips indicates the length of the discontinuity. If both tips exceed the rock walls height, the discontinuity is recorded as "through-going". Accordingly, discontinuities of which one or both ends are visible are categorized as "one end visible" or "both ends visible". On rare occasions, neither of both ends can be recorded, then the discontinuity is documented as "neither ends visible".

3. **Orientation:** The attitude of a discontinuity in space is called the orientation. The orientation is described by the dip direction (azimuth) and the dip of the line of the steepest declination in the plane of discontinuity.

4. **Roughness:** Roughness is the inherent alignment and waviness of the surface at the mean level of a discontinuity. A large-area waviness may also alter the dip locally. All roughness measurements were recorded manually. Therefore, we distinguish between smooth, slightly rough, and rough, both on mesoscopic and field scale.

5. **Aperture:** Aperture is the perpendicular distance between adjacent rock walls of a discontinuity where the space between them is filled with a material, that is, filling.

6. **Filling:** Filling is the material that separates the adjacent rock wall of a discontinuity and is usually weaker than the parent rock. In our study, typical filling materials are calcite, clay, debris, or quartz. These includes thin mineral coatings and healed discontinuities such as quartz and calcite veins.

In spite of the above-mentioned records, it must be emphasized again how important the calculation of the intersecting angle is. Due to an irregular outcrop wall, the distance between the starting point and the recorded discontinuity (i.e. apparent spacing) is measured incorrectly. This apparent distance distorts the true orientation of the discontinuity. The true angle $\delta$ between the scanline orientation (trend/plunge) and the normal to the recorded discontinuity was calculated according to

$$\delta = \arccos(\cos(\alpha_{\mathrm{n}} - \alpha_{\mathrm{s}})\cos(\beta_{\mathrm{n}})\cos(\beta_{\mathrm{s}}) + \sin(\beta_{\mathrm{n}})\sin(\beta_{\mathrm{s}})), \tag{1}$$

where $\alpha_s$, $\beta_s$, $\alpha_n$, and $\beta_n$ denote the trend direction of the tape line (scanline), the plunge direction of the tape line with regard to trend orientation, and the dip direction and dip of the normal to the fracture, respectively. Further, the true angle $\delta$ and the apparent spacing $\chi_{\mathrm{s}}$ between two discontinuities were used to calculate the true spacing $\chi_{\mathrm{r}} = \cos(\delta)\chi_{\mathrm{s}}$. By evaluating the ratio of true discontinuity spacing $\chi_{\mathrm{r}}$ and the number of discontinuity observations the mean discontinuity spacing $\bar{\chi} = \lambda^{-1} = (\sum \chi_{\mathrm{r}})n^{-1}$ was determined, where $\lambda$ denotes the discontinuity frequency. The evaluation of the mean discontinuity track length with respect to the required accuracy has already been discussed in detail by Priest and Hudson (1981). The authors have made various suggestions when the end of a discontinuity trace on an investigated rock wall may not be visible, for example, due to geological events or anthropogenic processes. In our study, the proposed solution for censored semi-trace length sampling was adopted to evaluate the mean discontinuity track length $\mu_i$. If the distribution of trace length over the recorded rock wall follows a negative exponential function (power law), then the frequency $f(l)$ is given by $f(l) = \lambda e^{-\lambda \bar{\chi}}$. Following the approach of Priest and Hudson (1981), for censored semi-trace length analysis the mean discontinuity trace





length was determined according to

$$\mu_i^{-1} = \mu^{-1} - (ce^{-\mu c})(1 - e^{-\mu c})^{-1}, \tag{2}$$

where $\mu$ and $c$ denote mean population frequency and concealed trace length of the recorded discontinuity, respectively. Typ-
ically, the concealed trace length is one order of magnitude smaller than the recorded trace length. Note that discontinuities
exceeding the height of the outcrop wall could be detected at the nearest level above and/or below the quarry.

The recorded discontinuities were evaluated with MATLAB (2018) after Hudson (2005) and Markovaara-Koivisto and Laine
(2012). Markovaara-Koivisto and Laine (2012) developed an open source MATLAB code for the visualization of scanline
survey results that was adapted to the purpose of this study. All measurements were recorded according to the metric system
of measurement.

### 2.4 Laboratory Measurements: Petrophysical Characterization of Samples

During the field work, loose rock blocks were taken in each quarry for further petrophysical characterization. Compacted
limestones, dolomites, and red, dolimitc rocks were named MKB (black Massenkalk), MKY (yellow Massenkalk), and MKR
(red Massenkalk), respectively. After sampling the rocks were named according to the sampled quarry, sampled working
level, rock type, and sample number. Representative, cylindrical cores with a diameter of $40\,\mathrm{mm}$ were extracted from each of
these blocks by diamond core drilling perpendicular to the bedding direction. In addition, it was possible for one rock sample
(HKW-2-MKB-2-S2) to be cored directly at an exposed rock wall in the outcrop of Hagen Hohenlimburg with a mobile drilling
machine. All samples, whether cored in the laboratory or the outcrop, were saw-cut plane-parallel and their end faces were
ground square to the maximal possible length $l$ (Table 4). A diameter-to-length ratio of 1:2.5 was aimed for, but this could not
always be achieved due to the high density of pre-existing fractures in some blocks. This target sample geometry is generally
recommended for triaxial deformation experiments on cylindrical samples (Paterson and Wong, 2005), a topic that exceeds the
scope of our current study. All steps of the preparation were conducted with water as coolant and for rinsing removed material.
Following preparation, samples were oven-dried at $60\,^\circ\mathrm{C}$ for about $48\,\mathrm{h}$. Basic petrophysical properties were determined on
identically prepared samples at ambient conditions, except permeability, which was derived under elevated pressures.
Bulk density $\rho_{\mathrm{geo}}$ was calculated from the geometrical volume of the cylindrical samples and their dry masses. Grain density
$\rho_{\mathrm{grain}}$ was gained from pycnometer measurements on rock powder, produced by crushing and grinding of leftover rock frag-
ments, in compliance with the German DIN 18124 standard. By evaluating the ratio between bulk and grain density, the total
porosity was determined according to $\phi_{\mathrm{tot}} = 1 - (\rho_{\mathrm{geo}}/\rho_{\mathrm{grain}})$. The connected porosity, that is, the externally accessible and
connected pore volume, was determined using the difference of the masses of dry and saturated samples with distilled water
(see Duda and Renner, 2013).

Ultrasonic P- and S-wave velocities, $v_\mathrm{P}$ and $v_\mathrm{S}$, were determined on dry and saturated samples from first-arrival measurements
using an ultrasound benchtop unit composed of a waveform generator, two identical broadband ultrasound sensors ($1\,\mathrm{MHz}$
centre frequency and $0.5\,\mathrm{in}$ diameter), and a digital storage oscilloscope ($200\,\mathrm{MHz}$ sampling frequency). Measurements were
performed parallel to the cylinder axis, that is, the drilling direction. Velocities were calculated by dividing the sample length





by the determined arrival times less the travel time in the assembly parts. Drained dynamic Young's modulus $\tilde{E}_\mathrm{d}$ and Poisson's ratio $\tilde{\nu}_\mathrm{d}$ were calculated by dry P- and S-wave velocities, $v_\mathrm{P,dry}$ and $v_\mathrm{S,dry}$, and bulk density assuming isotropy (see Mavko et al., 2020). Moreover, saturated P- and S-wave velocity, $v_\mathrm{P,sat}$ and $v_\mathrm{S,sat}$, and the density of the fluid-saturated sample was used to calculate undrained Poisson's ratio $\tilde{\nu}_\mathrm{ud}$ and Young's modulus $\tilde{E}_\mathrm{ud}$, assuming isotropy and employing Gassmann's hypothesis (Gassmann, 1951) that S-wave velocity remains unaffected by the presence of a fluid ($v_\mathrm{S,dry} = v_\mathrm{S,sat}$).

Thermal conductivity $\lambda_\mathrm{dry}$ of the dry sample cores was determined by a thermal conductivity scanner using an optical-scanning-method at ambient conditions (Popov, 1997). The scanner consists of an emitter and a measuring unit, that is moved along the sample at a fixed distance. The emitted light and heat radiation is focused on the surface of the sample, which heats up the sample pointwise. To ensure absolute absorption of the energy, a part of the sample, usually a strip, was painted black. Furthermore, it was ensured that the samples for thermal conductivity measurements meet the geometrical requirements for

sample size in order to reduce boundary effects. Infrared temperature sensors are located at a fixed distance from the emitter (lead sulfide infrared receiver) and measure the temperature difference of the sample before and after heating. The thermal conductivity was determined by comparison with known standards. In this study, we used the standards of quartz ($\lambda_\mathrm{qtz} = 1.35\,\mathrm{W(m\,K)^{-1}}$) and titanium alloy ($\lambda_\mathrm{Ti} = 6.05\,\mathrm{W(m\,K)^{-1}}$). According to the manufacturer (Lippmann and Rauen GbR; TCS No. 2010-013), the determination of thermal conductivity is subject to an absolute error of approximately $3\,\%$.

Permeability $k$ of the samples were determined using Darcy's Law (Darcy, 1856) and a conventional Hoek cell (Hoek and Brown, 1997) to apply axial load, pore-fluid pressure, and confining pressure. Axial pistons and the pre-saturated samples were jacketed by a rubber tube to prevent oil from penetrating the sample, that is, a connection between confining and pore pressure. In addition, confining and pore-fluid pressure were kept below the axial pressure. A hand pump was operated supplying the axial load ($12\,\mathrm{MPa}$). A computer-controlled, high-pressure metering pump was used to apply confining pressure on the sample

by compressing distilled water ($10\,\mathrm{MPa}$). Axial pore-fluid flow was ensured through central bores in the axial loading pistons. Distilled water was pumped from a water reservoir through the lower end face of the samples either by applying constant flow rates ($0.001$ to $0.15\,\mathrm{l\,h^{-1}}$) by a second, identical metering pump. The lower axial loading piston was equipped with an outlet pipe discharging fluid pressure to atmosphere (i.e. $1\,\mathrm{bar}$). The temperature and pressure of the pore-fluid was measured to calculate the temperature- and pressure-dependent fluid viscosity according to Wagner (2009). Only permeability of the

naturally fractured dolimite sample (HKW-2-MKB-2-S2) was investigated at three different stress stages. Each permeability of the samples indicated below corresponds to the arithmetic mean value of all experiments conducted with various flow rates for conditions of steady-state fluid flow. ‘

## 3   Results

In the following, the general findings from the field discontinuity investigations, which are found in all outcrops, are pre-

sented first, before each outcrop is described individually. Then, the results of the laboratory investigations are introduced. Furthermore, descriptive links between both field and laboratory results are established.





## 3.1 Field Discontinuity Observations

A total of 1068 discontinuity observations were recorded and classified (e.g., discontinuity type, filling, and roughness) by using the field surveys in the three outcrops in Wuppertal, Hagen Hohenlimburg, and Hönnetal (Table 2). It was found that essentially

three main sets of discontinuity orientations are predominant in all outcrops (Table 3). In Wuppertal the sets are grouped into the directions NNW−SSE, NE−SW, and NW−SE (Fig. 3c,d). In contrast, the discontinuity sets in Hagen Hohenlimburg and Hönnetal are oriented towards NNE−SSW, NW−SE and ENE−WSW, and N−S, W−E and ENE−WSW, respectively (Figs. 4c,d; 5c,d). The dip angles of the discontinuities were documented very steep, that is, between $80°$ and $90°$ (dip) in all three quarries.

**"Wuppertal"**. During the field observations in Wuppertal, scanline surveys were taken in two orientations NNW−SSE and NE−SW. Eight scanline surveys were measured with a total length of $66.6\,\mathrm{m}$ and 461 discontinuity measurements were performed (Table 3). We recorded steep dip angles for all joint orientations (Fig. 3). More than 130 measurements could be assigned to each of the three main discontinuity sets. Orientations of 257/88, 145/82, and 017/89 (dipdir/dip) were determined for the these three sets. A total of 129 discontinuities were recorded as open fractures, 110 as closed fractures, 113 as filled

fractures and 109 with slickensides along the fracture plane. Four discontinuities exceeded the height of the rock face, both ends were visible in 356 recordings, one end was recorded in 93 discontinuities, and in eight joints neither end was visible. We recorded 270 fillings within the discontinuities, 116 of which were calcite veins, 74 fractures were filled with recent debris, and 80 fractures had clay as filling material. The roughness recordings on field scale showed 354 smooth surfaces, 81 fractures were slightly rough and 26 fractures were recorded as rough. On mesoscopic scale, 159, 250, and 52 joints were recorded

as smooth, slightly rough, and rough, respectively. The mean discontinuity trace length and mean discontinuity spacing are $6.65\,\mathrm{m}$ and $0.15\,\mathrm{m}$, respectively (Fig. 3a,b).

**"Hagen Hohenlimburg"**. In Hagen Hohenlimburg the scanline surveys were conducted on two exposed rock walls with the orientations NNW−SSE and ENE−WSW. Over a total length of $76.15\,\mathrm{m}$ 361 discontinuities were recorded, which were divided into three sets of main orientations (Table 2). The set 1 lists 134 discontinuity observations, that show a mean common

orientation of 224/89 (dipdir/dip) (Table 3). Sets 2 and 3 comprise 95 and 132 discontinuities with an average orientation of 282/84 and 345/80 (dipdir/dip), respectively. The discontinuities recorded here show a varying angle of dip and no clear predominant dip and direction (Fig. 4). Of the 361 recorded discontinuities, 86 could be classified as open, 52 as closed, 156 as filled and 67 as slickenside discontinuities. Here, 20 discontinuities exceeded the height of the rock wall, in 300 recordings both ends were visible, one visible end was documented for 37 cases, and neither end was registered in four joints. Calcite is

listed as the dominant filling material with 160 entries, recent debris was recorded in 116 observations within the fractures, and clay was found in 31 discontinuities. The roughness of 194 discontinuities was classified as smooth, 116 as slightly rough and 51 as rough on the field scale. On the mesoscopic scale 77 smooth, 184 slightly rough and 100 rough discontinuities were observed. The mean joint track length and the mean joint distance differ significantly from the measurement results from the other quarries and amount to $1.23\,\mathrm{m}$ and $0.24\,\mathrm{m}$, respectively (Fig. 4a,b).



**"Hönnetal"**. In Hönnteal 246 discontinuities were recorded on two rock faces, which are oriented in the direction NNW−SSE
and ENE−WSW. All in all 62.76 m of scanline surveys were recorded and three sets of predominant orientations have been
identified (Table 2). The number of observations for each set was recorded at 82, 24, and 140, and the dominant orientations
for each set were derived at 266/75, 331/79, and 359/87 (dipdir/dip) (Fig. 5). 51 of 246 observations were made for open
discontinuities, 51 for closed, 121 for filled, and 23 fractures showed slickensides on their fracture surfaces. The height of the
rock face was exceeded by 29 discontinuities, both ends were visible in 116 joints, one end was found in 96 joints, and in
five joints neither end was visible. Calcite was discovered as the dominant filling material with 137 records, 45 showed debris
deposits and six discontinuities were filled by clay deposits. On a field scale, the majority of the fracture surfaces were classified
as smooth (216), whereas the number of slightly rough (26) and rough fracture surfaces (four) is relatively small. In contrast,
on a mesoscopic scale, 55, 155, and 36 discontinuities were found to have smooth, slightly rough, or rough fracture surfaces,
respectively. The mean discontinuity trace length was determined to be 3.94 m and lies between the mean trace lengths of the
other two outcrops. The mean discontinuity distance is similar to the one determined during the surveyed discontinuities in
Hagen Hohenlimburg and was determined to be 0.27 m.

### 3.2 Laboratory Characterizations

Samples could be obtained from all outcrops of dolomite and limestone, although it was challenging to obtain samples from
the Hönnetal due to the lack of revealed outcrops. Most of the exposed Hönnetal limestone rocks were pure and showed a very
low grade of dolomitization. Nevertheless, some dolomites have been sampled. Dolomite samples from Hagen Hohenlimburg
occurred as highly fractured rocks with karst and highly porous formations ranging from a few millimeters to several centime-
ters. In total, from the collected rock blocks of the Wuppertal, Hagen Hohenlimburg, and Hönnetal outcrops six, eight, and six
samples could be prepared, respectively. Furthermore, it was possible to drill a sample directly from the existing rock face in
Hagen Hohenlimburg, that is described separately below.

The mean bulk densities of the samples retrieved from the outcrops in Wuppertal, Hagen Hohenlimburg, and Hönnetal were
2704 ±39, 2537 ±128, and 2620 ±46 kg m$^{-3}$, respectively (Table 4). These mean values were determined independently of
whether they were limestone or dolomite. In Wuppertal as well as in Hagen Hohenlimbug, one dolomite sample was investi-
gated in each case, which differs significantly from the other samples and was not included in the calculation of the mean bulk
densities. Taking into account the measurement uncertainty and standard deviation, all samples examined showed comparable
densities. A similar trend was observed for the mean grain densities, as these values are also very close together. The mean
grain densities of the samples from Wupptertal, Hagen Hohenlimburg and Hönnetal amounted to 2801 ±51, 2777 ±41, and
2848 ±32 kg m$^{-3}$, respectively.

Laboratory measurements to determine the total and connected porosities indicated that the dolomites investigated were sig-
nificantly more porous than the limestones. This trend was evident at all quarries. The volumetric proportion of pores in the
dolomites was on average twice as high as in the limestones among almost all samples. Our uncertainty analysis showed that
no significant volume of isolated pores and pre-existing cracks could be determined in the investigated samples. Taking into
account the measurement uncertainties, the total and connected porosities overlapped (Fig. 6a).





Ultrasound P- and S-wave velocities of the dry samples ranged between 3000 and $6500\,\mathrm{m\,s^{-1}}$ and 1300 and $3500\,\mathrm{m\,s^{-1}}$,

respectively. Regardless of the outcrop, it was shown that the dry P-wave velocities of the limestones tended to be 1000 to $1500\,\mathrm{m\,s^{-1}}$ higher than those of the dolomites. The Wuppertal samples have shown that the dolomites and limestones cannot be distinguished according to the S-wave velocities. Both rock types had comparable S-wave velocities. In contrast, the samples of the remaining outcrops showed a clear difference in S-wave velocities between dolomite and limestone. In Hagen Hohenlimburg and Hönnetal the S-wave velocities of the limestones were about 500 to $1000\,\mathrm{m\,s^{-1}}$ higher than those of the

dolomites. These differences were also apparent in the drained dynamic Youngs' moduli. The mean drained Young's moduli of the limestone and the dolomites amounted to about 60 and $70\,\mathrm{GPa}$, respectively. The mean drained Poisson's ratio, that is, the relative relationship between P- and S-wave velocities, is about 0.35 for both rock types, taking into account the measurement uncertainties and the sample variability. Upon Saturation, P-wave velocities of the samples collected in Wuppertal, Hagen Hohenlimburg, and Hönnetal increased averagely by up to 4, 11, and $14\,\%$. This increase was also apparent in the undrained

Poisson's ratios. The undrained Young's moduli hardly differ from the drained ones. In this case, increases of 1 to $3\,\mathrm{GPa}$ were mainly recorded.

The results of the measurements of the thermal conductivities showed a scattering, with a slight tendency that the mean thermal conductivity of the limestones is at least as high as the one of the dolomites. The dolomite samples from Wuppertal were an exception, as these samples showed relatively high thermal conductivities that even exceeded those of the limestones. On

average, the highest and lowest conductivities could be determined on the Wuppertal and the Hagen Hohenlimburg samples, respectively. Measured thermal conductivities of all samples varied between 1.10 and $3.40\,\mathrm{W\,(m\,K)^{-1}}$.

The permeabilities measured correlated as expected with the corresponding porosities, that is, permeability increase with increasing proportion of connected pore volume (Fig. 6b; Table 4). The measured permeabilities of the dolomites from Hagen Hohemlimburg are the highest and averaged to about $4.27 \times 10^{-16}\,\mathrm{m^2}$. The Hagen Hohenlimburg limestones were on average

about three orders of magnitude less permeable than the limestones. The differences in permeability between dolomite and limestone samples amounted to approximately two orders of magnitude in Wuppertal, whereas dolomites and limestones in Hönnetal showed similar permeabilities of the same order of magnitude, that is, permeabilities in the range of $10^{-18}\,\mathrm{m^2}$. Only one sample from Hönnetal was low permeable (permeability of about $4 \times 10^{-20}\,\mathrm{m^2}$), but here the associated connected porosity was also significantly smaller than that of the comparative samples.

In the Hagen Hohenlimburg quarry, a healed vein in the limestone could be sampled directly in the outcrop. Before drilling, the orientations of the rock wall $f_1$ and the healed vein $f_2$ within the rock wall were recorded. The sample was drilled perpendicular to $f_1$ and parallel to $f_2$. The dip direction and angle of $f_1$ was 326° and 70° with respect to dip direction, respectively. In comparison, the measured dip orientation and angle of $f_2$ was 243° and 72° with respect to dip direction, respectively. During preparation, the cored sample was cut into two sections: S1 and S2. The laboratory investigations were carried out on the longer

sample S2. This sample had high porosities of about $10\,\%$. Based on our laboratory measurements, and given the uncertainties, no isolated pore space could be determined. The dry as well as the saturated ultrasound velocities could not be distinguished from the ones of the dolomite and limestone samples. Due to the saturation of the pore space with distilled water, an increase in the P-wave velocity of $10\,\%$ could be determined. Also, the thermal conductivity did not differ from the ones of the other





samples from Hagen Hohenlimburg. The determined Poisson's ratio was the largest in this study, whereas the Young's modulus
was in the lower range of all measured values. The permeability of this sample was determined under different pressures and a
strong pressure dependence was shown. The relatively high permeability decreased by about one and a half orders of magnitude
due to the minor increase of the stress state. At the highest applied pressure, however, the permeability also increased again by
half an order of magnitude. An additional X-ray microtomography scan of the sample showed that the fracture connects both
sample end faces but the fracture aperture exhibit large variations, that is, from open to closed.

## 4  Discussion

The discontinuities of deep carbonates in the Rhine-Ruhr area were characterized and described using outcrop scanline surveys.
Also the results of laboratory investigations of density, porosity, dynamic elastic moduli, thermal conductivity, and permeability
of the collected representative rock samples of the Devonian Reef Complex were presented. In the following, we correlate the
field and laboratory investigations and address the geothermal potential of these deep carbonates in the Rhine-Ruhr region. We
focus our results of the DFN on discontinuity length, density, orientation, and aperture. Characterizing the DFN requires an
access to the reservoir rock at reservoir related depth which is not present. However, we have reasonable concerns to expect
comparable discontinuity orientations in our geological subsurface model as in our studied outcrops (Lorenz et al., 1991; Narr,
1991). For example, Narr (1991, 1996) presented how to derive DFN from drill core samples of reservoirs.

### 4.1  Estimating the Geological Subsurface Model of the Carbonate Reservoir

The geological setting in the Rhine-Ruhr metropolitan area is highly complex, with details still being debated (e.g., Brix et al.,
1988). However, we can name large tectonic events, which have influenced local formations to a varying extent:

1. Crustal thinning, which enhanced the formation of a shelf sea during the Early Devonian (e.g., Dallmeyer et al., 2013).

2. An overall NW-movement that favoured reef deposits and is known today as the geothermal horizon of interest (e.g.,
   Franke et al., 2017).

3. The ongoing NW-movement, which changed the material deposits and thus finally created the Ruhr coal district during
   the Carboniferous (e.g., Meschede, 2018).

4. At least two different extension sequences during the Early Triassic and Early Cretaceous (Drozdzewski and Wrede,
   1994).

5. A NE-SW-directed shortening during the Late Cretaceous (e.g., Brix et al., 1988).

6. Further extensional regime since the Eocene (Kley, 2013).

Due to the complexity of the geological formation, our underground model has been simplified with respect to the most
important tectonic features such as folds and faults. The foundation of our model is an approx. 300 m thick carbonate layer,





dipping northwards at a shallow dip angle of about 30 to 40° (see Fig. 1; Jansen et al., 1986; Drozdzewski et al., 2007). We expect this layer to be at depths between 4000 to $6000\,\mathrm{m}$ (DEKORP Research Group, 1990). This layer forms the Devonian

basement, which is overlaid by interbedded sequences of sand, clay, silt, and coal layers from the Carboniferous period. These sediments can be mapped to the surface (e.g., Brix et al., 1988). Many folds and thrusts are found within these extremely complex interbedded sequences (for more details see Brix et al., 1988; Drozdzewski and Wrede, 1994); in this study, we will focus on the two most prominent fracture sets striking to NNW−SSE and NE−SW. Both fracture sets are also the dominant orientations in the Rhine-Ruhr area with respect to the world stress map (Heidbach et al., 2016). These previously documented

directions of the dominant fracture sets correspond very well with our results of the scanline measurements in the three different outcrops Wuppertal, Hagen Hohenlimburg, and Hönnetal (Table 3). Brudy et al. (1997) showed that no significant stress orientation changes have to be expected in the brittle crust with depth. Consequently, it is reasonable to assume that we can expect at least similar discontinuity directions in our target horizon of the deep Devonian limestone, that is, the potential geothermal reservoir. However, if the locations of the three investigated opencast mines are extrapolated in the direction of the

maximum horizontal stress (e.g., Heidbach et al., 2016), the carbonate reservoir is located in the area of Essen, Bochum, and Dortmund with respect to the simplified geological setting (Fig. 1). This assumption allows us to predict local DFN in the deep Devonian limestone (i.e. naturally fractured carbonate reservoir), whose exact depth and characteristics should be verified by additional geophysical prospecting techniques to further describe the geothermal potential of this reservoir (e.g., Hirschberg et al., 2015).

We carried out laboratory experiments under ambient and elevated pressure conditions to gain insights into the petrophysical properties of the potential reservoir. The derived porosities of the limestone samples are in agreement with literature values between 1 and $6\,\%$ (Fig. 6a; c.f. Gebrande, 1982). The porosity of carbonate rocks may increase by a broad variation of processes. Next to dissolution, cementation, and recrystallization the process of dolomitization is the most common one. Dolomitization describes the geochemical process of replacing $\mathrm{Ca}$ ions by $\mathrm{Mg}$ ions, forming dolomite from calcite: $2\mathrm{CaCO_3} +$

$\mathrm{Mg^{2+}} \rightarrow \mathrm{CaMg(CO_3)_2} + \mathrm{Ca^{2+}}$. Lucia et al. (2007) pointed out that dolomitization may increase the carbonates porosity by $13\,\%$. This correlates fairly well with our results (cf. Fig. 6a). Consequently, porosity measurements are in accordance with the outcome of the field study: porosity decreases towards core reef formation, in agreement with Homuth et al. (2015b).

The P- and S-wave velocities as well as the matrix permeabilities and dynamic mechanical properties derived in the laboratory provide statistical information for further numerical simulations of the reservoir. The thermal conductivity results are appropri-

ate for the analyzed rock formations (c.f. Čermák and Rybach, 1982; Clauser and Huenges, 2013; Jorand et al., 2015). The low connected porosities match with the determined permeabilities. Compacted limestone (Massenkalk), which is characterised by a low connected porosity, showed the lowest permeability, whereas the dolomite showed an inverse relation (see Fig. 6b). Although the results of the determined matrix permeability seem to be comparable with similar geothermal reservoirs (e.g., Homuth et al., 2015a), we do not expect sufficient reservoir permeability in 4000 and $6000\,\mathrm{m}$ depth. Therefore, we have fo-

cused on potential DFN within the geothermal reservoir, whose influence on fluid flow in fractured reservoirs has already been discussed in detail (e.g., Guerriero et al., 2013).



## 4.2 Evaluating a Discrete Fracture Network in the Carbonate Reservoir: From Outcrop and Laboratory Measurements to Implications for Fluid Flow on Reservoir Scale

The uncertainties associated with the description of naturally fractured systems and DFN are strongly dependent on sampling
effects such as truncation or censoring (Bonnet et al., 2001; Baecher and Christian, 2005). Basically, truncation effects explain
the influence of underestimating small characteristics due to the resolution of the sampling method, that is, outcrop and/or
survey size. Censoring effects occur when large characteristics are incompletely observed due to the outcrop size mentioned or
when characteristics are selected due to subjective choice (Priest and Hudson, 1981; Lacazette, 1991). While the trunctuation
effects ca be reduced by a sophisticated set of data and resolution, the censoring effect can be reduced depending on the
method applied, adapted technology, and the amount of data obtained (Santos et al., 2015). In the literature these uncertainties
are typically described as aleatory and epistemic (Der Kiureghian and Ditlevsen, 2009). Various survey methods and their
corrections with regard to the effects mentioned are described in more detail in Bonnet et al. (2001).

The approach we have chosen to use the scanline survey method described by Priest and Hudson (1981) is a widely accepted
method for the characterization of fracture networks (e.g., La Pointe and Hudson, 1985; Lacazette, 1991; Narr, 1996). The
advantages of the method are especially the acquisition time and the associated verification as well as the acquisition costs
compared to other methods (e.g., LIDAR outcrop survey; Wilson et al., 2011). The uncertainties due to scanline surveys are
generally known and have been critically questioned for decades (e.g., Terzaghi, 1965; Cruden, 1977; La Pointe and Hudson,
1985; Ganoulis, 2008). However, due to the long period of application, the method explained by Priest and Hudson (1981) has
been adapted for a variety of geological and outcrop settings, which leads to various error corrections regarding the application
case. This allows us to keep the limitations and sources of error as low as possible. We pursued the concerns about epistemic
uncertainties and identified the main problems in defining the locations of representative areas to evaluate a discrete fracture
network in the carbonate reservoir in the Rhine-Ruhr area.

### 4.2.1 Orientation of the Fracture Network

The architecture of each open pit mine studied is directly related to the directions of the dominant discontinuities, that is,
NNW−SSE and NE−SW (Fig. 2). Both rock faces are approximately perpendicular to each other in each quarry. The high
angle between the scanline tape and the recorded fracture sets allows us to neglect the blind spots, as discussed in Lacazette
(1991). In order to map all the features of the outcrop, we decided to carry out surveys at different levels of the quarry. The
main discontinuity orientations were documented as NNW−SSE, NW−SE, and NE−SW. These results fit with the general
observations documented in the literature (e.g., Brix et al., 1988). In fact, many of the discontinuities studied on the outcrop
scale can be related to residual stress and stress release during unloading regimes (Nickelsen and Hough, 1967; Roberts, 1974).
However, it was observed that the regional fracture sets were continuous and consistent in orientation over large areas of
several thousand square meters (Kelley and Clinton, 1960; Stearns and Friedman, 1972; Hancock and Bevan, 1987). When
comparing the recorded discontinuity orientations with the orientation of the maximum horizontal stress (Heidbach et al.,
2016), we propose to focus on discontinuities that are approximately oriented N−S for future shallow geothermal applications.





Pre-existing discontinuities oriented parallel to the maximum horizontal stress could be observed openly, while discontinuities oriented perpendicular to this stress tend to be closed (Lorenz et al., 1991). It can therefore be assumed that higher fracture permeability can be expected in the N−S direction.

### 4.2.2 Filling and Surface Roughness of the Fracture Network

Information about fracture fillings and surface roughness can be used to estimate the reservoir behaviour under in-situ con-
ditions, since fillings and roughness have a direct influence on the elastic, hydraulic and thermal properties of the reservoir. The observation of filling materials, like calcite, indicate paleo fluid flow paths which might be utilized by advanced drilling methods (e.g., hydraulic fracturing; Dahi Taleghani and Olson, 2013). Before utilizing, the material changes between host rock and vein material might be used to predict fracture orientations in depth by further geophysical driven studies (e.g., density changes, reflection coefficients). The presented results in this study can be used for a first impression of the petrophysical
properties, like P- and S-wave velocity, of the host rock itself. Further studies have to prove, if the material change between the encountered filling materials and the host rock is sufficient enough for seismic interpretation. More than half of all observations showed paleo filling materials like calcite or slickensides on the fracture surface, approx. $58\,\%$ and $17\,\%$, respectively. This results into $25\,\%$ of cracks which might develop during other tectonic origins. Those, unfilled cracks might be interpreted as open or closed in the subsurface. If the fractures appear to be open, fracture roughness is the next property which should be
considered in deriving a proper geothermal model of the area.

It is very complex to determine fluid flow by both numerical and analytical methods along two irregular faces, that is, natural fracture surface. However, the impact on fluid flow along rough fracture surfaces was already shown and discussed (e.g., Brown, 1987). In the scope of this study, fracture roughness was ascertained qualitatively regarding the applied method. The results indicate predominating smooth fracture surfaces on the field scale ($> 10^{-1}$ m) and mainly slightly rough fracture surfaces on
the mesoscopic scale ($\leqslant 10^{-1}$ m). In the literature roughness on field scale are typically described as waviness or straightness (ISRM, 1978). Verifying the determined roughness is only possible with difficulty. However, considering the numerous tectonic events and its repetitive reactivation of pre-existing fractures, smooth fractures tend to be reasonable.

### 4.2.3 Connectivity of the Fracture Network

The results of our scanline suveys revealed that discontinuities with short trace lengths occur predominantly on the field
scale. This observation was made in all three quarries. Almost $70\,\%$ of the recorded discontinuities classified as "both ends visible" tended to be between 1 and $2\,\mathrm{m}$ long. It is conceivable that the observed trace lengths increase with increasing outcrop length and height. The recorded end types suggest that $40\,\%$ of all discontinuities investigated exceeded the observed outcrop height. In other words: $40\,\%$ of all measured trace lengths might be underestimated by the method. This source of error was attempted to be minimized by the censored semi-trace length analysis according to Priest and Hudson (1981). Contrary, $60\,\%$
of all joints counted showed a start and end tip (Table 2). However, a full 3D discontinuity analysis of the entire quarry will likely reflect better trace length results depending on the accuracy. At this point, economic considerations must be made regarding the required survey accuracy and associated costs. Regarding our joint analysis from Wuppertal, approx. 70 % of





all observations are classified as short discontinuities (Table 3). In Hagen Hohenlimburg approx. $50\%$ of the observations are rather to be classified as short. In Hönnetal, approx. $47\%$ of all recorded joints had both ends determined. The comparison
of the discontinuity track lengths with the true discontinuity distances gives an idea of a connected connection network. With reference to the mean trace lengths and mean true discontinuity distances recorded in Wuppertal, Hagen Hohenlimburg, and Hönnetal, we expect comparable probabilities for connected joint networks in each outcrop. However, based on the observations of many karst formations and altered host rocks by hydrothermal veins in Hagen Hohenlimburg, we conclude that there is a higher permeability in the Hagen Hohenlimburg reservoir and in the surrounding areas than in the other studied ares. The
distribution of this reservoir in the subsurface is still under debatte (e.g., Salamon and Königshof, 2010) and its conditions due to karstification still have to be proven.

The densities of the dolomites and carbonates determined in the laboratory did not show significant differences, which was to be expected, but the comparatively high porosity of the dolomite samples indicated a moderate pore and crack volume. Consequently, on the one hand, it can be assumed that a higher density of potentially hydraulically stimulateable pores and
cracks (or fracture network) is found in the dolomite layers on the field scale. On the other hand, the comparable densities pose a warning regarding the unreflected interpretation of, for example, density measurements in naturally fractured carbonate reservoirs. Such measurements do not appear to be very useful for differentiating between low porous limestone and relatively high porous dolomite. The influence of pore and crack volume in the dolomite samples was also reflected in the measured ultrasonic velocities, as the dolomitic samples showed the lowest P-wave velocities. It is known that P-waves are generally
more sensitive to pore spaces and unfilled cracks than S-waves, which propagate only through the solid rock matrix (e.g., Mavko et al., 2020). The S-wave velocities of dolomites and carbonates hardly differ, which is in good agreement with the given densities. A warning must also be given at this point, as the healed vein sample showed almost identical ultrasonic velocities compared to the other samples (Table 5). The saturation of the healed vein sample resulted in the smallest increase in P-wave velocity. Therefore, it is important to observe the temporal and spatial variation of the elastic wave velocities during
a potential reservoir stimulation of the reservoir in order to get conclusions about its characteristics. In the next stage these findings can be of great importance for the interpretation of flow rates in the deep carbonate reservoir. Ahrens et al. (2018) demonstrated experimentally that there is an uniform correlation between P-wave velocity and the key parameter for transient flow processes, that is, hydraulic diffusivity, in the direction of fluid flow during inelastic deformation. Their observation allows the conclusion that monitoring variations in P-wave velocities in field surveys can indicate changing fluid flow rates in,
for example, evolving reservoirs. Consequently, these measurements bear the potential to give insights into deformation state, connectivity, and the hydraulic properties of pore and fracture networks in the reservoir.

The low impact of the fluid saturation on the dynamic moduli of the samples indicates that heterogeneous pore and/or crack networks prevail in the investigated samples. In the measuring or drilling direction of the samples only a small increase of the P-velocities and thus also of the undrained dynamic moduli occurs. This indicates that the connected pore and crack space
tends to be locally distributed (c.f. upper Hashin-Shtrikman bound for a simplified two phase carbonate material; Hashin and Shtrikman, 1963), since ultrasonic velocities only characterize the momentary elastic behavior along a wave path, which was chosen by the waves due to its above-average rigidity. On the scale of wavelength, the samples are thus probably heterogeneous.



Moreover, a velocity anisotropy should also prevail, which is particularly relevant for seismic surveys. However, a possibly existing heterogeneous and local distribution of the pore and crack volume has the advantage that it is more likely to encounter

large pores and long and/or wide open cracks than would be possible with a homogeneous porosity distribution. It can be assumed that the heterogeneity possibly favours the fluid flow.

The sampled healded vein from Hagen Hohenlimburg exhibited a major pressure dependence of permeability in the range of the applied pressures, that suggests a rather low permeability ($< 10^{-18}$) of the target horizon in the expected depth. By means of hydraulic stimulation, however, the fractures prevailing there and their networks could possibly be reactivated, thus

increasing the permeability. However, this prediction clearly depends on the orientation of the fracture network in the stress field and the fluid pressure. However, for geothermal projects it is indispensable to carry out a seismic risk before planning to manipulate the local stress conditions, that is, the reactivation of fractures.

## 5   Conclusion

We investigated the geothermal potential of a carbonate reservoir in the Rhine-Ruhr area, Germany, by combining outcrop

scanline surveys of pre-existing discontinuities and laboratory measurements of petrophysical properties on sample scale. The target horizon of interest is an approximately $300\,\mathrm{m}$ thick and widely distributed compacted limestone layer (Massenkalk) from Late Devonian in 4000 to $6000\,\mathrm{m}$ depth, dipping northwards at a shallow dip angle of about 30 to 40°. If we extrapolate the course of the exposed Devonian limestone layers, which were investigated at three outcrop analogues, in the direction of the maximum horizontal stress, the carbonate reservoir probably extends below Essen, Bochum, and Dortmund. Our petrophysical

laboratory measurements on representative outcrop samples show that there is insufficient matrix porosity, permeability, and thermal conductivity in the reservoir rocks. This indicates that the characterisation of discrete fracture networks could be the key to the successful implementation of deep geothermal projects in the Rhine-Ruhr metropolitan region.

At the examined outcrops located within the Devonian Reef Complex, our scanline surveys revealed three main discontinuity orientations within the compacted limestones (NNW−SSW, NW−SE, and NE−SW) with dipping angles between 80 and 90°.

These discontinuity sets were in very good agreement with previously documented ones, that are the predominant orientations in the Rhine-Ruhr area with respect to the world stress map. With a simplified assumption that the stress state does not change significantly with depth, it can be assumed that at least one of these fracture sets can be expected in the target horizon, that is, a naturally fractured carbonate reservoir. We propose to focus on discontinuities that are approximately N−S oriented for upcoming geothermal applications, considering that this direction is parallel to the maximum horizontal stress. Hence, possibly

opened fractures with higher permeability could be expected here.

The results of our comprehensive scanline surveys also provided information about the possible filling and surface roughness, which are of significant importance for estimating the fluid flow in the reservoir. More than half of all observations showed paleo-filling materials such as calcite or slickensides on the fracture surface. The measured fracture sets showed mainly smooth and slightly rough fracture surfaces on the field scale and the mesoscopic scale, respectively. Considering the numerous tectonic

events in the Rhine-Ruhr region and the resulting repeated reactivation of pre-existing fractures, slightly smooth fracture



surfaces are more likely to occur in the reservoir. Discontinuities sets with short trace lengths occurred predominantly in all three outcrops. Almost $70\,\%$ of the recorded discontinuities classified as "both ends visible" tended to be between $1$ and $2\,\mathrm{m}$ long. We compared the mean trace lengths with the mean true discontinuity spacing in order to get an idea of a coherent fracture network in the reservoir. Taking all measurements into account, we expect comparable probabilities for interconnecting

fracture networks in the deep reservoir. However, based on the observations of many karst formations and altered host rocks by hydrothermal veins in Hagen Hohenlimburg, we conclude that the permeability is greater in the regions of the Devonian reef facies than in the other areas investigated. This assumption was also confirmed by our laboratory measurements, whose results suggest the presence of heterogeneities that may favours fluid flow.

Our presented results may provide the basis for improved subsurface characterization with respect to the structure and char-
acteristics of the naturally fractured Devonian limestone reservoir in the Rhine-Ruhr area. More sophisticated geophysical prospecting techniques and a combination of pressure and temperature dependent laboratory measurements have to elaborated to further verify the reservoir depth, orientation as well as its fracture density and connectivity. Further studies have to prove, if the fractures prevailing could possibly be reactivated and/or connected to fracture network, thus enhancing fluid flow in the carbonate reservoir. In addition to further discontinuity properties, the local stress field should be further verified in follow-up
studies.The understanding of the structural characteristics of the fracture network and their impact on the elastic, thermal, and hydraulic properties of the reservoir that in turn defines the geothermal potential will benefit from, for instance, seismic surveys and in-situ measurements in a pilot borehole.





**Figure 1.** Simplified geological map of the Rhine-Ruhr metropolitan area (Germany) and three cross-sections of the Remscheid-Altena anticline (modified after Jansen et al., 1986; Drozdzewski et al., 2007). Here, three stone pits have been chosen for field surveys: Osterholz (Wuppertal), Oege (Hagen), and Asbeck (Hönnetal), which are part of the Devonian Reef Complex (Massenkalk).





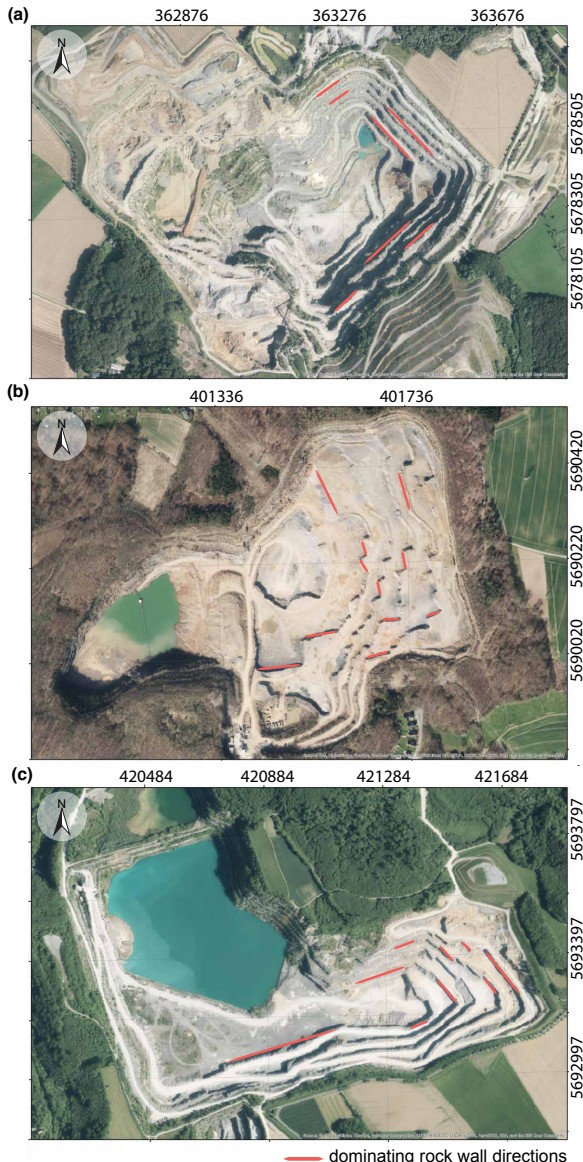

— dominating rock wall directions

**Figure 2.** Map view of all three study areas in which scanline surveys were carried out: (a) The Osterholz stone pit, which was operated by Kalkwerke H. Oetelshofen GmbH & Co. KG in Wuppertal. (b) The Oege stone pit operated by Hohenlimburger Kalkwerke GmbH in Hagen Hohenlimburg. (c) The Asbeck rock mine managed by Lhoist Deutschland, Rheinkalk GmbH in Hönnetal. Red lines indicate the predominant rock walls, that is, the direction of mining. In each of the three outcrops there were two dominant mining directions, which were approximately perpendicular to each other. This figure was created using ArcGIS® software by Esri (Basemap, World Imagery: http://www.arcgis.com/home/item.html?id=10df2279f9684e4a9f6a7f08febac2a9). ArcGIS© and ArcMap™ are the intellectual property of Esri and are used herein under license. Copyright © Esri. All rights reserved. For more information about Esri™ software, please visit www.esri.com.





**Table 1.** Overview of the documented discontinuity properties and their classification as defined for scanline surveys according to Markovaara-Koivisto and Laine (2012). The roughness of the discontinuities was classified on field scale ($> 10^{-1}$ m) and on mesoscopic scale ($\leqslant 10^{-1}$ m).

| property | class | | | | |
|---|---|---|---|---|---|
| | 1 | 2 | 3 | 4 | 5 |
| discontinuity type | open | closed | filled | slickensides | dyke |
| ending type | through going | both ends visible | one end visible | neither ends visible | — |
| filling | calcite | debris | quartz | clay | — |
| roughness on field scale | smooth | slightly rough | rough | — | — |
| roughness on mesoscopic scale | smooth | slightly rough | rough | — | — |





**Table 2.** Overview of the recorded properties and their classification of the measured discontinuities in the three studied outcrops in Wuppertal, Hagen Hohenlimburg, and Hönnetal. See text and Table 1 for a detailed description of the classes.

| outcrop | property | class | | | | | total |
|---|---|---|---|---|---|---|---|
| | | 1 | 2 | 3 | 4 | 5 | |
| Wuppertal | discontinuity type | 129 | 110 | 113 | 109 | — | 461 |
| | ending type | 4 | 356 | 93 | 8 | — | 461 |
| | filling | 116 | 74 | — | 80 | — | 270 |
| | roughness at field scale | 354 | 81 | 26 | — | — | 461 |
| | roughness at mesoscopic scale | 159 | 250 | 52 | — | — | 461 |
| Hagen Hohenlimburg | discontinuity type | 86 | 52 | 156 | 67 | — | 361 |
| | ending type | 20 | 300 | 37 | 4 | — | 361 |
| | filling | 160 | 67 | — | 31 | — | 258 |
| | roughness at field scale | 194 | 116 | 51 | — | — | 361 |
| | roughness at mesoscopic scale | 77 | 184 | 100 | — | — | 361 |
| Hönnetal | discontinuity type | 51 | 51 | 121 | 23 | — | 246 |
| | ending type | 29 | 116 | 96 | 5 | — | 246 |
| | filling | 137 | 45 | — | 6 | — | 188 |
| | roughness at field scale | 216 | 26 | 4 | — | — | 246 |
| | roughness at mesoscopic scale | 55 | 155 | 36 | — | — | 246 |
| all outcrops | discontinuity type | 266 | 213 | 390 | 199 | — | 1068 |
| | ending type | 53 | 772 | 226 | 17 | — | 1068 |
| | filling | 413 | 186 | — | 117 | — | 716 |
| | roughness at field scale | 764 | 223 | 81 | — | — | 1068 |
| | roughness at mesoscopic scale | 291 | 589 | 188 | — | — | 1068 |





**Table 3.** Overview of the settings and the results of the scanline surveys performed on rock walls in the outcrops Wuppertal (WOH), Hagen Hohenlimburg (HKW), and Hönnetal (HLO). The abbreviations MKY, MKB, and MKR correspond to the coloured rock types Massenkalk yellow (dolomite), Massenkalk black (limestone), and Massenkalk red (dolomite), respectively. Three main families or sets of discontinuity orientations were found to be predominant in all outcrops.

| scanline survey | $l_{\text{scan}}$ | $n$ | $\bar{l}_{\text{dis}}$ | $\bar{h}_{\text{dis}}$ | $\bar{d}_{\text{dis}}$ | discontinuity orientation (dipdir/dip) | | |
| | (m) | (-) | (m) | (m) | (m) | set 1 | set 2 | set 3 |
|---|---|---|---|---|---|---|---|---|
| WOH-2-MKB-1 | 9.70 | 74 | 1.63 | < 0.01 | 0.13 | 329/89 | 237/84 | – |
| WOH-3-MKB-1 | 12.22 | 84 | 1.64 | < 0.01 | 0.15 | 259/87 | 190/87 | – |
| WOH-4-MKB-1 | 14.10 | 58 | 2.31 | < 0.01 | 0.25 | 265/85 | 150/86 | – |
| WOH-4-MKB-2 | 8.10 | 39 | 0.92 | < 0.01 | 0.20 | 146/70 | 193/81 | 270/88 |
| WOH-4-MKY-1 | 5.92 | 81 | 3.14 | < 0.01 | 0.08 | 247/89 | 198/88 | – |
| WOH-4-MKY-2 | 3.94 | 44 | 0.56 | < 0.01 | 0.09 | 261/84 | 203/65 | – |
| WOH-5-MKB-1 | 10.10 | 59 | 0.87 | < 0.01 | 0.20 | 259/87 | 135/77 | – |
| WOH-5-MKB-2 | 2.52 | 22 | 0.85 | < 0.01 | 0.11 | 268/83 | 313/69 | 038/69 |
| HKW-2-MKB-1 | 17.40 | 41 | 2.75 | 0.02 | 0.43 | 240/87 | – | – |
| HKW-2-MKB-2 | 9.90 | 42 | 3.58 | 0.03 | 0.24 | 333/71 | – | – |
| HKW-3-MKB-1 | 5.80 | 36 | 0.77 | < 0.01 | 0.19 | 290/89 | 348/70 | – |
| HKW-3-MKB-2 | 4.75 | 30 | 0.54 | < 0.01 | 0.22 | 194/71 | – | – |
| HKW-1-MKY-1 | 17.30 | 97 | 0.58 | < 0.01 | 0.21 | 207/83 | – | – |
| HKW-4-MKB-1 | 6.80 | 40 | 0.54 | < 0.01 | 0.20 | 277/87 | 175/81 | – |
| HKW-5-MKB-1 | 7.43 | 14 | 0.76 | < 0.01 | 0.28 | 212/82 | – | – |
| HKW-5-MKY-4 | 6.80 | 46 | 0.23 | < 0.01 | 0.36 | 351/86 | 284/82 | 212/82 |
| HLO-4-MKB-1 | 13.66 | 50 | 3.31 | < 0.01 | 0.30 | 351/75 | – | – |
| HLO-4-MKB-2 | 10.00 | 49 | 2.32 | < 0.01 | 0.21 | 176/77 | – | – |
| HLO-6-MKB-1 | 17.70 | 34 | 6.77 | 0.06 | 0.53 | 254/87 | 018/86 | – |
| HLO-6-MKB-2 | 10.80 | 30 | 3.55 | 0.01 | 0.40 | 253/81 | 134/74 | 195/86 |
| HLO-6-MKB-3 | 10.60 | 84 | 4.67 | 0.02 | 0.14 | 271/89 | – | – |

$l_{\text{scan}}$: scanline length; $n$: number of discontinuities; $\bar{l}_{\text{dis}}$: mean trace length; $\bar{h}_{\text{dis}}$: mean discontinuity aperture; $\bar{d}_{\text{dis}}$: mean discontinuity spacing. The resolution of the length measurements amounts to 0.01 m.





**Figure 3.** Results of all scanline investigations at the quarry in Wuppertal: (a) discontinuity trace length diagram, (b) discontinuity distance diagram, (c) and (d) stereogram and rose diagram of measured discontinuity sets.





**Figure 4.** Results of all scanline investigations at the quarry in Hagen Hohenlimburg: (a) discontinuity trace length diagram, (b) discontinuity distance diagram, (c) and (d) stereogram and rose diagram of measured discontinuity sets.





**Figure 5.** Results of all scanline investigations at the quarry in Hönnetal: (a) discontinuity trace length diagram, (b) discontinuity distance diagram, (c) and (d) stereogram and rose diagram of measured discontinuity sets.





**Table 4.** Matrix properties, permeabilities, and thermal conductivities as determined by laboratory measurements on limestone and dolomite samples from Wuppertal (WOH), Hagen Hohemlimburg (HKW), and Hönnetal (HLO). The abbreviations MKY, MKB, and MKR indicate the colored rock types Massenkalk yellow (dolomite), Massenkalk black (limestone), and Massenkalk red (dolomite), respectively.

| sample | $l$ (mm) | $\rho_{\text{geo}}$ (kg m$^{-3}$) | $\rho_{\text{grain}}$ (kg m$^{-3}$) | $\phi_{\text{tot}}$ (%) | $\phi_{\text{con}}$ (%) | $k$ (m$^2$) | $\lambda_{\text{dry}}$ (W (m K)$^{-1}$) |
|---|---|---|---|---|---|---|---|
| WOH-4-MKY-3 A | 62.28 | $2695 \pm 7$ | $2801 \pm 1$ | $4.1 \pm 1.8$ | $4.1 \pm 0.7$ | $(2.90 \pm 0.10) \times 10^{-18}$ | $2.64 \pm 0.08$ |
| WOH-4-MKY-3 B | 83.99 | $2787 \pm 7$ | $2830 \pm 1$ | $1.5 \pm 1.8$ | $6.0 \pm 1.0$ | $(1.80 \pm 0.07) \times 10^{-18}$ | $3.02 \pm 0.09$ |
| WOH-4-MKY-3 C | 55.08 | $1755 \pm 5$ | – | – | $2.5 \pm 0.4$ | $(5.30 \pm 0.20) \times 10^{-17}$ | $3.33 \pm 0.09$ |
| WOH-4-MKB-1 A | 83.53 | $2706 \pm 7$ | $2712 \pm 1$ | $0.2 \pm 1.8$ | $1.7 \pm 0.3$ | $(1.90 \pm 0.07) \times 10^{-20}$ | $2.62 \pm 0.08$ |
| WOH-4-MKB-1 B | 82.05 | $2698 \pm 7$ | $2801 \pm 1$ | $3.7 \pm 1.8$ | $1.7 \pm 0.3$ | $(1.60 \pm 0.06) \times 10^{-19}$ | $2.63 \pm 0.08$ |
| WOH-4-MKB-1 C | 82.91 | $2704 \pm 7$ | – | – | $1.7 \pm 0.3$ | $(1.90 \pm 0.07) \times 10^{-20}$ | $2.63 \pm 0.08$ |
| HKW-4-MKR-1 A | 29.18 | $2558 \pm 8$ | $2771 \pm 1$ | $7.7 \pm 1.8$ | $7.1 \pm 1.2$ | $(2.20 \pm 0.09) \times 10^{-16}$ | $2.18 \pm 0.06$ |
| HKW-4-MKR-1 B | 41.59 | $2420 \pm 7$ | $2779 \pm 1$ | $12.9 \pm 1.8$ | $14.0 \pm 2.3$ | $(1.00 \pm 0.04) \times 10^{-15}$ | $3.36 \pm 0.10$ |
| HKW-4-MKR-1 C | 39.31 | $2018 \pm 6$ | – | – | $14.7 \pm 2.4$ | $(7.60 \pm 0.32) \times 10^{-16}$ | $2.38 \pm 0.07$ |
| HKW-4-MKY-2 A | 39.39 | $2389 \pm 7$ | $2770 \pm 1$ | $13.8 \pm 1.8$ | $10.7 \pm 1.8$ | $(2.40 \pm 0.10) \times 10^{-17}$ | $2.19 \pm 0.06$ |
| HKW-4-MKY-2 B | 34.13 | $2420 \pm 7$ | $2768 \pm 1$ | $12.9 \pm 1.8$ | $10.3 \pm 1.7$ | $(1.30 \pm 0.05) \times 10^{-16}$ | $2.13 \pm 0.06$ |
| HKW-5-MKB-1 A | 83.89 | $2682 \pm 7$ | $2777 \pm 1$ | $3.4 \pm 1.8$ | $1.8 \pm 0.3$ | $(4.90 \pm 0.20) \times 10^{-20}$ | $2.62 \pm 0.08$ |
| HKW-5-MKB-1 B | 74.30 | $2660 \pm 7$ | $2772 \pm 1$ | $4.0 \pm 1.8$ | $2.7 \pm 0.4$ | $(1.40 \pm 0.06) \times 10^{-18}$ | $2.56 \pm 0.08$ |
| HKW-5-MKB-1 C | 76.92 | $2707 \pm 7$ | – | – | $1.3 \pm 0.2$ | $(2.40 \pm 0.10) \times 10^{-19}$ | $2.52 \pm 0.08$ |
| HKW-2-MKB-2-S2 | 92.25 | $2516 \pm 7$ | $2777 \pm 0.2$ | $9.4 \pm 1.8$ | $9.8 \pm 1.6$ | see Table 6 | $3.00 \pm 0.09$ |
| HLO-6-MKY-1 A | 80.43 | $2541 \pm 7$ | $2820 \pm 1$ | $9.9 \pm 1.8$ | $4.6 \pm 0.8$ | $(6.60 \pm 0.26) \times 10^{-18}$ | $2.33 \pm 0.07$ |
| HLO-6-MKY-1 B | 57.01 | $2621 \pm 7$ | $2875 \pm 1$ | $8.8 \pm 1.7$ | $5.6 \pm 0.9$ | $(3.30 \pm 0.13) \times 10^{-18}$ | $2.45 \pm 0.07$ |
| HLO-6-MKY-1 C | 58.05 | $2611 \pm 5$ | – | – | $5.6 \pm 0.9$ | $(4.20 \pm 0.17) \times 10^{-18}$ | $2.15 \pm 0.06$ |
| HLO-6-MKB-3 A | 81.32 | $2682 \pm 7$ | $2820 \pm 1$ | $4.9 \pm 1.8$ | $2.4 \pm 0.4$ | $(3.80 \pm 0.15) \times 10^{-20}$ | $2.67 \pm 0.08$ |
| HLO-6-MKB-3 B | 47.16 | $2643 \pm 7$ | $2875 \pm 1$ | $8.1 \pm 1.7$ | $4.3 \pm 0.7$ | $(2.60 \pm 0.10) \times 10^{-18}$ | $2.71 \pm 0.08$ |
| HLO-6-MKB-3 C | 41.53 | $2619 \pm 7$ | – | – | $3.7 \pm 0.6$ | $(2.20 \pm 0.09) \times 10^{-18}$ | $2.60 \pm 0.08$ |

$l$: sample length; $\rho_{\text{geo}}$: bulk density; $\rho_{\text{grain}}$: grain density; $\phi_{\text{tot}}$: total porosity; $\phi_{\text{con}}$: connected porosity; $k$: permeability; $\lambda_{\text{dry}}$: thermal conductivity. Quoted uncertainties reflect accuracy of the measurements.





**Table 5.** Ultrasound velocities and dynamic elastic moduli derived by laboratory measurements for limestone and dolomite samples from Wuppertal (WOH), Hagen Hohemlimburg (HKW), and Hönnetal (HLO). The abbreviations MKY, MKB, and MKR indicate the colored rock types Massenkalk yellow (dolomite), Massenkalk black (limestone), and Massenkalk red (dolomite), respectively.

| sample | $v_{P,dry}$ $(\mathrm{m\,s^{-1}})$ | $v_{S,dry}$ $(\mathrm{m\,s^{-1}})$ | $v_{P,wet}$ $(\mathrm{m\,s^{-1}})$ | $\tilde{\nu}_d$ $(-)$ | $\tilde{\nu}_{ud}$ $(-)$ | $\tilde{E}_d$ (GPa) | $\tilde{E}_{ud}$ (GPa) |
|---|---|---|---|---|---|---|---|
| WOH-4-MKY-3 A | $5660 \pm 20$ | $2990 \pm 200$ | $6420 \pm 30$ | $0.31 \pm 0.13$ | $0.36 \pm 0.09$ | $63 \pm 8$ | $65 \pm 6$ |
| WOH-4-MKY-3 B | $5590 \pm 20$ | $2850 \pm 140$ | $5800 \pm 20$ | $0.30 \pm 0.10$ | $0.34 \pm 0.07$ | $62 \pm 6$ | $63 \pm 5$ |
| WOH-4-MKY-3 C | $5790 \pm 40$ | $2929 \pm 340$ | $6086 \pm 40$ | $0.33 \pm 0.13$ | $0.35 \pm 0.1$ | $61 \pm 8$ | $62 \pm 7$ |
| WOH-4-MKB-1 A | $6430 \pm 20$ | $3120 \pm 170$ | $6470 \pm 20$ | $0.34 \pm 0.08$ | $0.35 \pm 0.08$ | $71 \pm 6$ | $71 \pm 6$ |
| WOH-4-MKB-1 B | $6310 \pm 20$ | $3030 \pm 160$ | $6520 \pm 20$ | $0.35 \pm 0.08$ | $0.36 \pm 0.07$ | $67 \pm 5$ | $67 \pm 5$ |
| WOH-4-MKB-1 C | $6380 \pm 20$ | $2800 \pm 140$ | $6480 \pm 20$ | $0.38 \pm 0.06$ | $0.38 \pm 0.05$ | $59 \pm 3$ | $59 \pm 3$ |
| HKW-4-MKR-1 A | $4050 \pm 30$ | $1740 \pm 150$ | $5580 \pm 50$ | $0.39 \pm 0.09$ | $0.45 \pm 0.05$ | $21 \pm 2$ | $22 \pm 1$ |
| HKW-4-MKR-1 B | $2970 \pm 10$ | $1350 \pm 60$ | $4330 \pm 20$ | $0.37 \pm 0.06$ | $0.44 \pm 0.03$ | $12 \pm 1$ | $13 \pm 1$ |
| HKW-4-MKR-1 C | $4050 \pm 20$ | $2230 \pm 180$ | $6050 \pm 40$ | $0.28 \pm 0.18$ | $0.42 \pm 0.07$ | $26 \pm 5$ | $29 \pm 2$ |
| HKW-4-MKY-2 A | $4530 \pm 20$ | $2460 \pm 220$ | $5430 \pm 30$ | $0.29 \pm 0.19$ | $0.37 \pm 0.1$ | $37 \pm 7$ | $40 \pm 5$ |
| HKW-4-MKY-2 B | $4810 \pm 30$ | $2570 \pm 280$ | $5210 \pm 40$ | $0.30 \pm 0.21$ | $0.34 \pm 0.17$ | $41 \pm 9$ | $43 \pm 7$ |
| HKW-5-MKB-1 A | $6450 \pm 20$ | $3320 \pm 190$ | $6560 \pm 20$ | $0.32 \pm 0.10$ | $0.33 \pm 0.09$ | $78 \pm 8$ | $78 \pm 8$ |
| HKW-5-MKB-1 B | $6190 \pm 20$ | $3380 \pm 220$ | $6350 \pm 20$ | $0.29 \pm 0.14$ | $0.30 \pm 0.12$ | $78 \pm 11$ | $79 \pm 10$ |
| HKW-5-MKB-1 C | $6410 \pm 20$ | $2770 \pm 140$ | $7120 \pm 30$ | $0.39 \pm 0.06$ | $0.41 \pm 0.05$ | $57 \pm 3$ | $58 \pm 3$ |
| HKW-2-MKB-2-S2 | $5125 \pm 13$ | $2150 \pm 71$ | $5600 \pm 15$ | $0.39 \pm 0.03$ | $0.42 \pm 0.03$ | $32 \pm 1$ | $33 \pm 1$ |
| HLO-6-MKY-1 A | $5027 \pm 14$ | $2540 \pm 115$ | $6440 \pm 20$ | $0.33 \pm 0.07$ | $0.41 \pm 0.04$ | $43 \pm 3$ | $46 \pm 2$ |
| HLO-6-MKY-1 B | $5183 \pm 20$ | $2530 \pm 160$ | $6200 \pm 30$ | $0.34 \pm 0.10$ | $0.40 \pm 0.06$ | $45 \pm 5$ | $47 \pm 3$ |
| HLO-6-MKY-1 C | $3628 \pm 10$ | $2419 \pm 140$ | $5930 \pm 30$ | $0.10 \pm 0.27$ | $0.40 \pm 0.06$ | $34 \pm 9$ | $43 \pm 2$ |
| HLO-6-MKB-3 A | $6255 \pm 20$ | $3120 \pm 170$ | $6450 \pm 20$ | $0.34 \pm 0.09$ | $0.35 \pm 0.08$ | $74 \pm 6$ | $74 \pm 6$ |
| HLO-6-MKB-3 B | $5822 \pm 32$ | $3210 \pm 300$ | $6200 \pm 40$ | $0.28 \pm 0.21$ | $0.31 \pm 0.17$ | $70 \pm 15$ | $72 \pm 13$ |
| HLO-6-MKB-3 C | $5849 \pm 40$ | $3500 \pm 400$ | $6290 \pm 40$ | $0.22 \pm 0.35$ | $0.28 \pm 0.27$ | $78 \pm 28$ | $82 \pm 23$ |

$v_{P,dry}$, $v_{P,wet}$: P-wave velocity of dry and wet samples; $v_{S,dry}$: S-wave velocity of dry samples; $\tilde{\nu}_d$: dynamic drained Poission's ratio; $\tilde{\nu}_{ud}$: dynamic undrained Poission's ratio; $\tilde{E}_d$: dynamic drained Young's modulus; $\tilde{E}_{ud}$: dynamic undrained Young's modulus. Quoted uncertainties reflect accuracy of the measurements.





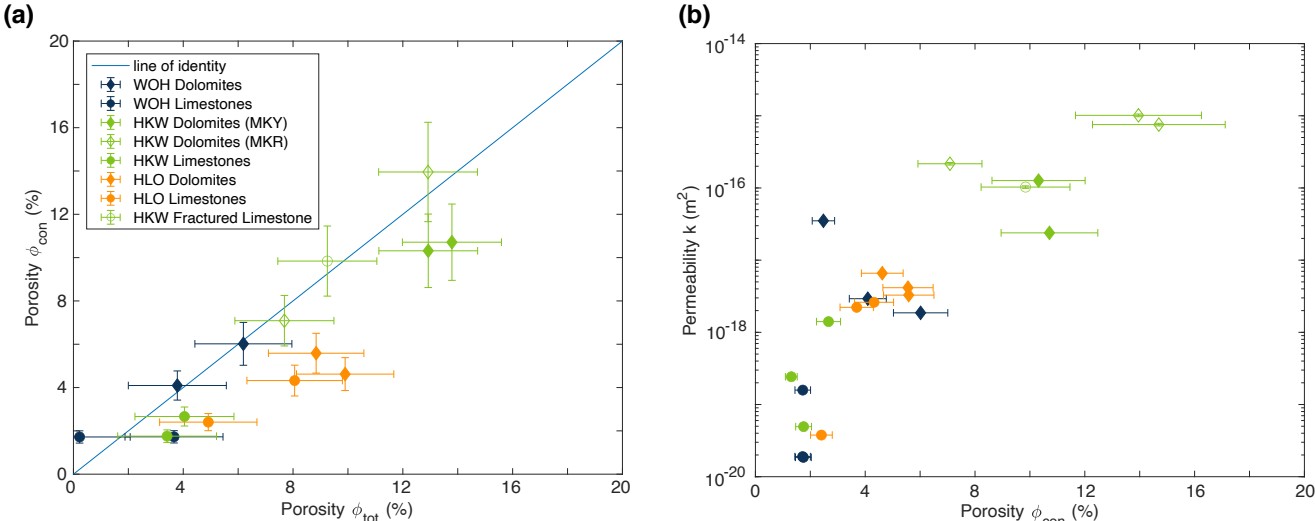

**Figure 6.** Correlation between (a) permeability and connected porosity and correlation between (b) connected and total porosity. Filling and coloring of markers indicate sample and outcrop, respectively. Measurement uncertainties are indicated by error bars. Where error bars do not exceed marker size, uncertainties appear to be small. The blue line in (b) indicates identity, that is, connected is equal to total porosity.





**Table 6.** Permeability of sample HKW-2-MKB-2-S2 as derived at different stress states. This limestone sample (MKB: Massenkalk black) was cored directly from a healed vein at an exposed limestone wall in the outcrop of Hagen Hohenlimburg (HKW).

| sample | axial load (MPa) | confining pressure (MPa) | $k$ (m$^2$) |
|---|---|---|---|
| | 7 | 5 | $(4.7 \pm 0.2) \times 10^{-16}$ |
| HKW-2-MKB-2-S2 | 12 | 10 | $(1.0 \pm 0.4) \times 10^{-18}$ |
| | 17 | 15 | $(5.9 \pm 0.2) \times 10^{-18}$ |

$k$: permeability; Quoted uncertainties reflect accuracy of the measurements.





*Acknowledgements.* We thank all students and researcher involved in field and laboratory measurements, in particular Mathias Nehler. MB, BA, and EHS acknowledge generous funding by the Federal Ministry of Education and Research for the project 3D-RuhrMarie ("FHpro-fUnt2016"). KL thanks the "Ministerium für Innovation, Wissenschaft und Forschung des Landes Nordrhein-Westfalen" for funding of the project GREENER ("FH ZEIT für FORSCHUNG").

*Financial support.* This research has been funded by the 3D-RuhrMarie project ("FHprofUnt2016"), ComLabgo project ("FH ZEIT für FORSCHUNG"), and GREENER project ("FH ZEIT für FORSCHUNG").

*Competing interests.* The authors declare that they have no conflict of interest.

*Data availability.* The measured values and results recorded within the study are available on reasonable request from the corresponding author.

*Author contributions.* Four authors have contributed to this paper. MB was the lead author and carried out the field investigations, laboratory measurements, and analytical calculations. BA helped with the laboratory measurements on transport and elastic properties. KL was involved in the field investigations. EHS provided helpful background knowledge. All authors were involved in the data interpretation and writing of the manuscript.





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
