# Peer review of "Characterization of fractures in potential reservoir rocks for geothermal applications in the Rhine-Ruhr metropolitan area (Germany)"

_Solid Earth, 2020_

## Short Comment (SC1) · 26 May 2020

In their paper, Characterization of fractures in potential reservoir rocks for geothermal applications in the Rhine-Ruhr metropolitan area (Germany), Dr. Balcewicz and co-authors document natural fractures in limestones exposed in quarries some tens of km to the south of the metropolitan areas of Essen, Bochum, and Dortmund, with the goal of extrapolating what kind of natural fracture patterns might be present within the same layers underneath those cities. This is an important topic because fractures do commonly control subsurface permeability, and would be critical to the viability of

the limestones as geothermal energy reservoirs. Moreover, a large amount of data has been carefully collected and presented in the paper, so I think it is potentially publishable in Solid Earth, and would particularly be a good contribution to the special issue on crustal fracturing.

However, I do have a few concerns with the paper as written. Despite a great wealth of data presented, there are some confusing and perhaps contradictory statements given which should be clarified. These mostly concern the orientation of the fractures in the quarries, and their inferred orientations in the subsurface. As well, the conclusions are fairly scant and qualitative, especially in terms of what type of fracture patterns are predicted to be in the subsurface and why. Lastly, and relatedly, I think the authors have the data to weigh in on some long-standing debates about fractures at the surface versus those in the subsurface, the implications of which have important consequences for the hypothetical geothermal capacity. I think to make their study broadly applicable to similar problems elsewhere, and thus to warrant publication in the scientific literature, the authors need to be firmer in their conclusions and explain the implications for their findings on natural fracture patterns in general.

To begin with, the outcrop data show a variety of fracture orientations, and the conclusions are ambiguous about what orientations we should expect in the subsurface. There appear to be three fairly clearly defined (by strike) fracture sets at Wuppertal, two (and perhaps a third) at Hönnetal, and not much systematic strike organization at Hagen Hohenlimburg (judging by the rose diagrams in Figures 3, 4, and 5). Importantly, the fracture strikes in the three quarries do not match one another. Is this because bedding is in a different orientation, accounting for a clockwise rotation at Hönnetal? It is not clear. If so, it may mean the fractures pre-date the folds. The best way to show that is to restore bedding and see what the fracture orientations do. But furthermore, the authors note a correspondence between the observed fracture orientations and the World Stress Map, but that map, in western Germany, has SHmax indicators in many directions. I would agree that NNW is probably the dominant one, but I see several

trending NE, WNW, and NNE. So the wide scatter in the outcrop data, combined with that in the map, makes questionable any correspondence between the two. And in any case, it is claimed that the dominant fracture orientations in quarry exposures are NNW and NE, and I suggest that is only true for one of the three quarries (Wuppertal).

Again, in Hönnetal it is unclear whether a third fracture set is present, judging by orientation; we might be able to objectively discriminate sets based on filling, but it is not clear which orientations are filled with what mineral. Fill type is an excellent way to discriminate between fracture relative ages and origins, and of course will have large implications on permeability. As well, partial cementation is a way to keep fractures open in the subsurface, regardless of the current state of stress (Laubach et al., EPSL, 2004).

Assuming the authors are correct and more-or-less N-S striking fractures would dominate the subsurface permeability, can we extrapolate any key parameters of the fracture patterns (length distributions, connectivity, intensity) based on the quarry exposures? These are only discussed very qualitatively. Are the N-S (or thereabouts) striking fractures more porous? Entirely sealed?

The quarry fractures dip 80 to 90 degrees, and yet the beds dip 30 to 40 degrees toward north. This implies a distinct non-orthogonality between the fractures and the beds. Is this true of all fractures of all fill types? All else being equal, I would assume opening mode fractures would form with either a vertical or horizontal attitude. Does the near-vertical attitude of fractures, and the significant dip of beds, mean that the fractures post-date the folding of the beds? Is that true of mineral-filled fractures (perhaps earlier) and clay- or debris-filled fractures (perhaps later-formed)? If so, presumably my above conjecture that the fractures pre-date the folds is wrong. Why then do the strikes in different quarries not match up? It could be any number of reasons (different stress states, different timings) but with a more thorough description of the observations, particularly documenting fracture orientation versus fill type, we could make more sense of the variation of the pattern from quarry to quarry. This is seemingly a prerequisite to

understanding the variation from quarries to subsurface.

Line-by-line comments:

14, also 567. Can omit "shallow" because that is a relative term, and you have already quantified it.

18. See general comment above: why do you focus on N-S fractures and not the other orientations? Especially since the stress map has a variety of SHmax orientations. Is your inferred insufficience of permeability ameliorated any if more fractures are present in the subsurface?

20-1. Is karstification only related to the facies fabric, or does it have anything to do with fracturing? Is there dissolution porosity associated with fractures?

37. "can be" for "be can"

73. "right-handed side" awkward/un-specific

127-9. Please clarify: there is a 150 m thick "carbonate layer" on top of dolomitic carbonates (which or course are also carbonates. Then the limestone beds have 1-5m thickness? The carbonate layer is composed of limestone beds? What are the bed boundaries like–is there fracture stratigraphy?

143. Again, awkward distinction between "dolomite" and "carbonate"–dolomite is a carbonate mineral.

200. Can you describe your method of establishing/categorizing the roughness more? How did you observe them at two scales? It's fine to just have three qualitative categories of roughness, but please take this opportunity to describe the roughness more. Is it imparted by stylolitization, branching, hooking?

204. Are any fractures filled by quartz? Looks like zero in Table 2-3.

213. What is the "true" spacing between two fractures that are not parallel? Better to

ust use the apparent spacing?

220. A negative exponential function is not, I don't think, the same as a power law. Exponential is $y = ab\hat{\ }x$ whereas power law is $y = ax\hat{\ }b$.

408. Why does tectonic motion favor the growth of reefs?

417, also 566. Why is the modeled layer 300 m thick? Why not 150, as in Line 127? Without quantifying predicted fracture attributes, what is the purpose of specifying a layer thickness?

429-31. I don't understand this sentence.

480. Unloading stress regimes–yes, highly possible. To determine whether this occurred, some discernment of fracture orientation versus fill would be useful.

487. This is very approximate usage of orientations when discussing potential for openness. Is the difference between "NNW" and N-S" significant?

497. Slickensides a filling material?

534-5: P-waves more sensitive to pore spaces and unfilled cracks? I thought it was precisely because S-waves only travel through the solid rock that S-waves are more sensitive.

589. "comparable"—any two numbers are comparable; you mean similar? How similar? Can you quantify intersection probabilities, if I drilled a hole at Dortmund?

---

## Referee Comment (RC1) · John Hooker (Referee) · 30 May 2020

Hello,

I apologize for the confusion, but I submitted my review for the paper, Characterization of fractures in potential reservoir rocks for geothermal applications in the Rhine-Ruhr metropolitan area (Germany), by M. Balcewicz et al., under another Copernicus ID. So you will find my detailed comments there.

Overall I think it is a solid and high quality effort that will make a good contribution to

the volume. But I do think there are fairly substantial revisions to be made, including clarifications about orientations, more quantitative conclusions, and broadening of the significance. This is why after much deliberation I went with major revisions rather than minor.

I thank the authors and editors for inviting me to review this very intriguing and important study.

Sincerely,

John Hooker

---

## Referee Comment (RC2) · Sadegh Karimpouli (Referee) · 16 Jul 2020

Dear Editor

The paper is about evaluation of potential geothermal reservoirs using field and lab data. The topic is high demanding and interesting for a broad range of communities. Data were surveyed and measured accurately, which worth to be published. The conclusions are qualitative, which goes back to the nature of data. I think more data from extensive field survey, surface sampling and core drilling to digital imaging, numerical

computation and simulation of, for example, DFN are needed to finalize this subject and to present a conceptual model. However, I support the paper for publication in this step, since it can be the basis for future advancements in this topic. The paper is well organized with perfect literature survey and proper English. However, it suffers from lack of visualization. Although the authors explained everything in detail in the main body, a reader tends to touch the results visually by figures and plots. So my main comment is about a more informative geological map showing the results of the study. I have also some minor comments which are as follow:

- Geological map lacks of enough data. I mean, all information in the text must be transferred into the geological map. Then, in a regional point of view, all data could be connected, judged and concluded. The most important points are:

i. Putting rose diagrams of all sites on the geological map

ii. Showing strike and dip of the target layers on each site and if possible on the other areas.

ii. Showing orientation of the dominant and/or present day maximum stress (either from literature or the authors observation).

I tried it as the attache figure. Based on maps in my figure, Hönnetal site is located on an anticline axis. The authors should be careful about combining fractures orientation data from this site with other sites. They should explain:

a. How are those rose diagrams are connected to each other?

b. How are they connected to the regional tectonic regime?

c. Line 417: "The foundation of our model is an approx. 300m thick carbonate layer, dipping northwards at a shallow dip angle of about 30 to 40 °". Fracture orientations on the anticline axis (HLO) show a different pattern compared to downward limbs (WHO, HKW). How do the authors combine them together?

- Section 4.2.1: The authors explain "The main discontinuity orientations were documented as NNW-SSE, NW-SE, and NE-SW." and then conclude "we propose to focus on discontinuities that are approximately oriented N-S for future shallow geothermal applications.". Is N-S one of the main directions or what? Is this conclusion on the basis of maximum stress direction? How is the contribution of the other factors such as fracture filling, conductivity and so on?

- Use different parameter for thermal connectivity and discontinuity frequency (both of them are $\lambda$)

- Figure 6: How do you translate connected pores more than total pores?
* * *
[Figure]

**Fig. 1.** An example of preferred geological map with data and results of this study

---

## Author Comment (AC1) · 11 Aug 2020

**Reviewer 1 (John Hooker)**

**John Hooker:** *"These mostly concern the orientation of the fractures in the quarries, and their inferred orientations in the subsurface. As well, the conclusions are fairly scant and qualitative, especially in terms of what type of fracture patterns are predicted to be in the subsurface and why. Lastly, and relatedly, I think the authors have the data to weigh in on some long-standing debates about fractures at the surface versus those in the subsurface, the implications of which have important consequences for the hypothetical geothermal capacity. I think to make their study broadly applicable to similar problems elsewhere, and thus to warrant publication in the scientific literature, the authors need to be firmer in their conclusions and explain the implications for their findings on natural fracture patterns in general."*

**Authors' reply:** We agree with you that the results of our fracture orientations were not sufficiently clear in the original version. Thank you for pointing out that you consider our study an important contribution to the long-standing debate on fractures on the surface compared to those underground. Nevertheless, we believe that our results are not universally applicable to all other geothermal sites due to the complex geological structures. However, our data and findings are particularly interesting for geothermal case studies that will also be conducted in the vicinity of an anticline. This is an important finding from the investigation of the fractures orientation in relation to the outcrops along the regional Remscheid-Altena anticline. By including the additional distinction between fracture orientation and fracture filling, we were able to formulate our conclusion more clearly (please see conclusion section 5). Furthermore, we now distinguish more clearly between the individual outcrops.

**John Hooker:** *"Importantly, the fracture strikes in the three quarries do not match one another. Is this because bedding is in a different orientation, accounting for a clockwise rotation at Hönnetal? It is not clear. If so, it may mean the fractures pre-date the folds. The best way to show that is to restore bedding and see what the fracture orientations do."*

**Authors' reply:** Thank you for that important remark, which mentions a point that we must emphasise more dominantly in the text. Yes, the fracture strikes in the three quarries do not match one another. We have the same explanation as you for this observation, namely that these differences are due to the formation of the anticline. We have now added this to the text (line 469):

> **"**All studied outcrops are located in the large scaled fold formation Remscheid-Altena anticline. However, there is a disagreement between the three outcrop results which might be an effect of the formation of the regional Remscheid-Altena anticline, different stress states, or different time of origin (Table 3). Due to the anticline formation, the strike directions of the present fractures in this region exhibit a rotation from the northern limb (Wuppertal) towards the tip of the anticline (Hönnetal). Fractures striking NE–SW are highly related to folding mechanism and are parallel oriented to fold axes which have been studied within the Rhine-Ruhr area (Drozdzewski, 1985; Brix *et al.*, 1988; DEKORP Research Group, 1990; Drozdzewski and Wrede, 1994)."

Besides, we have added this paragraph to refer more precisely to the chronological development of the fractures in Hagen Hohenlimburg and Hönnetal (line 480):

> "The observed strong scattering of the fracture strike directions in the dolomitic carbonates exposed in Hagen Hohenlimburg is due to their formation by hydrothermal veins during the Hercynian Orogney (Gillhaus et al., 2003). Furthermore, Gillhaus et al. (2003) explain that the existing NNW–SSE striking fractures are of post-Hercynian Orogeny origin. The cause of the slightly different fault strike directions in Hönnetal cannot be clearly specified according to the current state of scientific knowledge. Most likely, the fracture formation can be explained by various local and temporal stress anomalies and different formation times."

The study of chronological bedding and fracture formations and its restoration is a widely studied topic with different approaches. In many cases these investigations show a very simplified model based on single formations (e.g., Maerten and Maerten, AAPG Bull., 2006; Caumon *et al.*, Math. Geosci., 2009). Unfortunately, we could not sufficiently address this point, because it exceeds the scope of our study. But we would like to take up this point in the following studies, that are already scheduled to be published by the end of this year (for more details see Gonzalez de Lucio *et al.*, EGU General Assembly Conference Abstracts, 2020; Kruszewski *et al.*, EGU General Assembly Conference Abstracts, 2020).

**John Hooker:** *"But furthermore, the authors note a correspondence between the observed fracture orientations and the World Stress Map, but that map, in western Germany, has SHmax indicators in many directions. I would agree that NNW is probably the dominant one, but I see several trending NE, WNW, and NNE. So the wide scatter in the outcrop data, combined with that in the map, makes questionable any correspondence between the two. And in any case, it is claimed that the dominant fracture orientations in quarry exposures are NNW and NE, and I suggest that is only true for one of the three quarries (Wuppertal)."*

**Authors' reply:** Thank you for pointing out that our conclusion is not sufficiently explained. We have adapted our reasoning and added another important reference (Rummel & Weber, 1993). The new paragraph about the geological subsurface model of the carbonate reservoir, the predominant stress directions in western Germany, and the influence of the prominent Remscheid-Altena anticline reads as (line 475):

> "The dominant fracture strike directions NNW–SSE in Wuppertal agree with the structure of the regional Remscheid-Altena anticline (Fig. 1b) and the overall assumed mean principle stress direction according to the World Stress Map (Heidbach *et al.*, 2016) and additional available stress data (Rummel and Weber, 1993). In western Germany, or to be more precise in North Rhine-Westphalia, the World Stress Map contains a wide variability of mean principle stress directions (Heidbach *et al.*, 2016), that can be explained by shallow stress measurements, local anomalies which can be attributed to weak coal-seems, or regional NE–SW thrusts."

**John Hooker:** *"Again, in Hönnetal it is unclear whether a third fracture set is present, judging by orientation; we might be able to objectively discriminate sets based on filling, but it is not clear which orientations are filled with what mineral. Fill type is an excellent way to discriminate between fracture relative ages and origins, and of course will have large implications on permeability. As well, partial cementation is a way to keep fractures open in the subsurface, regardless of the current state of stress (Laubach et al., EPSL, 2004)."*

**Authors' reply:** We consider your commentary to be very appropriate and have accordingly added another figure to the manuscript (see Fig. 7). The new Figure 7 shows rose diagrams of all measured discontinuity sets as a function of their fracture filling, that is, whether the fractures are filled with (a) calcite or (b) debris. From our point of view, Figure 7 emphasizes the idea of focusing on discontinuities that are oriented towards NNW-SSE. Figure 7 shows that the NNW-SSE-striking faults are mainly filled with calcite. It can be found, that debris filled fractures do not show a distinct strike direction. In the text we refer to the figure as follows (e.g., line 540):

> "In addition, we present fracture orientations versus filling materials, these are, calcite or debris. The orientation of the recorded veins allows us to conclude, that many of the discontinuities studied on the outcrop scale can be related to residual stress and stress release during unloading regimes (cf. Nickelsen and Hough, 1967; Roberts,1974, Fig. 7)."

The quoted paper was very helpful and appropriate. Based on Laubach et al (2004), we expect at least partially open fractures in the direction of NNW-SSE, independent of the current stress state (line 550):

"In addition to the relative orientation of the fractures to the direction of the main principal stress, the filling is also decisive for whether the fractures are potentially open or closed in the subsurface (Laubach *et al.*, 2004). Thus, there might be open fractures that are not necessarily aligned in the direction of the principal stress and are still open. This is particularly true for fractures that are filled, for example, with cement (Laubach *et al.*, 2004)."

**John Hooker:** *"Assuming the authors are correct and more-or-less N-S striking fractures would dominate the subsurface permeability, can we extrapolate any key parameters of the fracture patterns (length distributions, connectivity, intensity) based on the quarry exposures? These are only discussed very qualitatively. Are the N-S (or thereabouts) striking fractures more porous? Entirely sealed?"*

**Authors' reply:** Yes, it is true that our results are primarily discussed qualitatively. But this can be explained by a number of reasons. First of all, the present manuscript is a preliminary study on which a comprehensive geomechanical model and further intensive laboratory measurements will be based. The aim of this manuscript is that we characterize the complex geological conditions and existing fracture orientations at the surface. In order to fulfill the requirements of a comprehensive publication, we explicitly address how potential underground reservoir conditions could be addressed by near-surface geological features in North Rhine-Westphalia, more precisely in the Rhine-Ruhr area. These investigations have not been covered to date for this highly complex geological environment. We expect that by answering the further comments, the question about the specificity of the N-S striking fractures has now been clarified in detail. N-S striking fractures are mainly calcite-filled. However, if we look at discontinuities that are oriented parallel to the main normal stress direction and exhibit a slightly different striking from N-S striking calcite-filled fractures, we assume that this open fractures may significantly compensate for the poor matrix permeability in the reservoir rock. Nevertheless, you are absolutely right that with the results in this manuscript we can only state an upper or possible lower limit regarding the fracture network and the associated fracture permeability. For this reason the present sentence in line 492is decisive:

„This assumption allows us to predict local DFN in the deep Devonian limestone (i.e., naturally fractured carbonate reservoir), whose exact depth and characteristics should be verified by additional geophysical prospecting techniques to further describe the geothermal potential of this reservoir (e.g., Hirschberg *et al.*, 2015)."

**John Hooker:** *"The quarry fractures dip 80 to 90 degrees, and yet the beds dip 30 to 40 degrees toward north. This implies a distinct non-orthogonality between the fractures and the beds. Is this true of all fractures of all fill types? All else being equal, I would assume opening mode fractures would form with either a vertical or horizontal attitude. Does the near-vertical attitude of fractures, and the significant dip of beds, mean that the fractures post-date the folding of the beds? Is that true of mineral-filled fractures (perhaps earlier) and clay- or debris-filled fractures (perhaps later-formed)? If so, presumably my above conjecture that the fractures pre-date the folds is wrong. Why then do the strikes in different quarries not match up? It could be any number of reasons (different stress states, different timings) but with a more thorough description of the observations, particularly documenting fracture orientation versus fill type, we could make more sense of the variation of the pattern from quarry to quarry. This is seemingly a prerequisite to understanding the variation from quarries to subsurface."*

**Authors' reply:** Yes, the observation you made applies to all fractures of all filling types. To answer this basic question of whether the fractures occurred after the beds were folded, we have included a new reference in the manuscript (see Gillhaus *et al.*, 2003). In addition, we have visualized the different strike directions of mineral-filled and debris-filled fractures in Figure 7. This helped us considerably to explain the variation from each quarry. In the results section 3.1 we have added the following sentences (line 311):

"By further differentiating the strike directions of the classified discontinuity sets further according to their filling material differences between calcite and debris-filled fractures can

be identified (Fig. 7). Our data tend to show a slight strike rotation between paleo-filled and debris-filled discontinuities in NNW–SSE direction."

We have added the following information to the description of the results of the Wuppertal outcrop (line 320):

"No significant difference in the strike direction of the paleo and debris-filled discontinuities can be detected (Fig. 7a,b). The average paleo-filled and debris-filled fracture strike directions are 177°and 178°, respectively."

The description of the results of the Hagen Hohenlimburg outcrop was extended by two sentences (line 342):

"Here, the disagreement between the average calcite-filled fracture strikedirection and that with debris becomes more apparent (Fig. 7c,d). The average paleo-filled and debris-filled fracture strike directions are 135° and 154°, respectively."

For the Hönnetal digestion we have summarized the results of the filling material differentiation in one sentence (line 361):

"The difference between the striking of calcite and debris-filled fractures is about 34° (Fig. 7e,f)."

In the discussion section 4.2.2 we discuss the new results of the filling material differentiation (line 565):

"The observed slight fracture rotation between paleo and recent or debris-filled discontinuities strongly suggests that the fill material may serve as an indication of their different tectonic origins. It is also remarkably that the average orientation of the recently filled cracks of the set 1 corresponds very well with the orientations of the main principal stress according to the World Stress Map (Heidbach *et al.*, 2016) and additionally available stress data (Rummel and Weber, 1993). The unfilled fractures might be interpreted as open or closed in the subsurface. If the fractures appear to be open, fracture roughness is the next property which should be considered in deriving a proper geothermal model of the area."

**Line-by-line comments:**

**John Hooker:** *"14, also 567. Can omit "shallow" because that is a relative term, and you have already quantified it."*
**Authors' reply:** Thank you, we adjusted the sentence accordingly and removed the word "shallow".

**John Hooker:** *"18. See general comment above: why do you focus on N-S fractures and not the other orientations? Especially since the stress map has a variety of SHmax orientations. Is your inferred insufficience of permeability ameliorated any if more fractures are present in the subsurface?"*
**Authors' reply:** On re-reading our sentence, we realized that it was not formulated precisely enough. We thank you for bringing this to our attention. The corrected sentences read (line 18):

"We proposed to focus on prominent discontinuities striking in NNW-SSE for upcoming geothermal applications, as these (1) are the most common, (2) strike in the direction of the main principal stress, (3) the fracture permeability significantly exceeds that of the reservoir rock matrix, and (4) because some of them are filled. Hence, the filled fractures bear the potential to be reopened by, for example, hydrochemical dissolution to create even better fluid efficiencies."

**John Hooker:** *"20-1. Is karstification only related to the facies fabric, or does it have anything to do with fracturing? Is there dissolution porosity associated with fractures?"*
**Authors' reply:** Thank you for the hint. In fact, we have recorded karstification related to hydrothermal process in Hagen Hohenlimburg. The corrected sentence reads (line 22):

"Our results indicate that even higher permeability can be expected for karstified formations related to the reef facies and hydrothermal processes."

**John Hooker:** *"38. "can be" for "be can"."*
**Authors' reply:** Thank you very much. We have corrected the sentence accordingly.

**John Hooker:** *"75. "right-handed side" awkward/un-specific."*
**Authors' reply:** You're absolutely right. We've improved the expression: "Three outcrop analogues on the eastern side of the northern Rhenohercynian Massif have been chosen for field survey and sample collection."

**John Hooker:** *"127-9. Please clarify: there is a 150 m thick "carbonate layer" on top of dolomitic carbonates (which of course are also carbonates. Then the limestone beds have 1-5m thickness? The carbonate layer is composed of limestone beds? What are the bed boundaries like – is there fracture stratigraphy?"*
**Authors' reply:** After reading it several times it became clear that the noted sentences are not conclusive. Many thanks for the hint. We have now clarified the description (line 129):
> "The studied limestone layers with a mean thickness of 1 to 5 m showed grayish, compacted layers of well-bedded carbonates with corals, stromatoporoidea, and bioclastic materials. This approximately 150 m thick horizon sits on top of dolomitic carbonates and is either related to the brachiopoda- and coral rich series of stromatopora series. The limestone bed boundaries show mechanical and fracture stratigraphy."

**John Hooker:** *"145. Again, awkward distinction between "dolomite" and "carbonate"–dolomite is a carbonate mineral."*
**Authors' reply:** Very good point. Accordingly, we have introduced the term "dolomitic carbonate" to distinguish it more clearly from the mineral "dolomite". We have made this change for the whole document.

**John Hooker:** *"200. Can you describe your method of establishing/categorizing the roughness more? How did you observe them at two scales? It's fine to just have three qualitative categories of roughness, but please take this opportunity to describe the roughness more. Is it imparted by stylolitization, branching, hooking?"*
**Authors' reply:** Thank you for the comment. We are happy to explain our qualitative measuring technique to determine fracture roughness on the meso- and field scale. We added the following sentence (line 207):
> "On the mesoscopic scale, the fracture surfaces were analysed for mineral steps, stylotites, or plumose structures. On the field scale, the fracture roughness was determined by wavelength measurements with a tape line."

**John Hooker:** *"213. Are any fractures filled by quartz? Looks like zero in Table 2-3."*
**Authors' reply:** Your observation is correct. We have not observed any quartz fillings and have accordingly deleted this term from the manuscript. This was a textual relic. The corrected sentence in line 208 reads:
> "In our study, typical filling materials are calcite, clay or debris."

**John Hooker:** *"213. What is the "true" spacing between two fractures that are not parallel? Better to ust use the apparent spacing?"*
**Authors' reply:** It is important to mention that we classify the recorded fracture networks as spatially homogeneous. Therefore, it is also possible to calculate the true spacing from the apparent spacing. You are absolutely right that in case of non-parallel fractures this calculation is not correct. For this reason, we have included the following sentence in the text (line 183):

"Based on our field observations, we make the simplified assumption that spatially homogeneous fracture networks are dominant in the outcrops, whereby this depends on the spatial scale of observation. In regions with spatially heterogeneous fracture networks, however, more complex methods than the scanline sampling technique are required for fracture characterization (Watkins *et al.*, 2015)."

We have also adjusted the sentence in line 221:
"Further, the true angle $\delta$ and the apparent spacing $\chi_s$ between two discontinuities were used to calculate the true spacing assuming a homogeneous fracture pattern $\chi_r = \cos(\delta)\,\chi_s$."

**John Hooker:** *"229. A negative exponential function is not, I don't think, the same as a power law. Exponential is y = abˆx whereas power law is y = axˆb."*
**Authors' reply:** You're absolutely right. This is a power-law relationship and we have adapted the sentence accordingly. Thank you for the remark.

**John Hooker:** *"408. Why does tectonic motion favor the growth of reefs?"*
**Authors' reply:** Thanks. In fact, the sentence was not particularly revealing. For a better understanding we have expanded this sentence, provided more detailed information, and added an additional reference (line 452):
"An overall NW-movement reduced clastic sedimentation within the sea and enabled the formation of reef carbonates on the clastic shelf. Beside those clastic shelf carbonates, other reef deposits formed on volcanic mounds within the hemipelagic realm and is the geothermal horizon of interest in this study (Franke *et al.*, 2017; Salamon & Königshof, 2010)."

**John Hooker:** *"417, also 566. Why is the modeled layer 300 m thick? Why not 150, as in Line 127? Without quantifying predicted fracture attributes, what is the purpose of specifying a layer thickness?"*
**Authors' reply:** Thanks, a mistake has crept in here. The modeled layer is 150 m and not 300 m thick. The error is due to the fact that in the literature the term carbonate horizons refers to the limestones and dolomitic layers studied (see Jansen *et al.*, 1986; Drozdzewski *et al.*, 2007). The complex of limestones and dolomitic carbonates is about 300 m thick. In the investigated outcrops, however, only the first 150 m, that is, the limestone layers, were exposed. Based on the previous investigations (e.g. DEKORP Research Group, 1990), we mention the layer thickness to bundle all information on the target horizon. We corrected the remarked all text passages accordingly.

**John Hooker:** *"429-31. I don't understand this sentence."*
**Authors' reply:** With this sentence we would like to state that if we extrapolate the carbonate layers exposed in the quarries in their direction of incidence, these carbonates are located approximately below the cities of Essen, Bochum, and Dortmund. These deep carbonates, which are located below these three cities, represent the target reservoir of this study. This extrapolation of the carbonates in the direction of incidence of the layers corresponds in this case approximately to the direction of the main principal stress (Heidbach *et al.*, 2016; Rummel & Weber, 1993). It should be noted that this extrapolation is based on the simplified geological model (Fig. 1a). We have like to adapt the sentence, so that our statement is now more understandable (line 487):
"However, if the carbonate layers exposed in the investigated opencast mines are extrapolated in dip direction of bedding, the carbonate reservoir of interest is approximately located below the cities Essen, Bochum, and Dortmund at depth of 4000 to 6000 m (Fig. 2). This extrapolation of the carbonates in dip direction corresponds in this case approximately to the direction of the main principal stress (Heidbach *et al.*, 2016; Rummel and Weber, 1993) with respect to the simplified geological setting (Fig. 1a)."

**John Hooker:** *"480. Unloading stress regimes–yes, highly possible. To determine whether this occurred, some discernment of fracture orientation versus fill would be useful."*

**Authors' reply:** Thank you for pointing out that we need to address the different filling material in order to support our assumption. Therefore, a new Figure 7, which shows two rose diagrams with calcite or debris filling, can be found accordingly (line 540):

> "In addition, we present fracture orientations versus filling materials, these are, calcite or debris. The orientation of the recorded veins allows us to conclude, that many of the discontinuities studied on the outcrop scale can be related to residual stress and stress release during unloading regimes (cf. Nickelsen and Hough, 1967; Roberts, 1974; Fig. 7)."

The orientation of the calcite-filled fractures supports our hypothesis that the NNW-SSE-striking fractures are particularly interesting for future geothermal applications. (line 545):

> "When comparing the recorded discontinuity orientations with the orientation of the maximum horizontalstress (Heidbach *et al.*, 2016), we propose to focus on discontinuities that are oriented NNW-SSE for future geothermal applications, which are highly probable filled by calcite."

**John Hooker:** *"487. This is very approximate usage of orientations when discussing potential for openness. Is the difference between "NNW" and N-S" significant?"*

**Authors' reply:** Due to the general agreement with you that the fracture strike orientation N-S does not reflect the measured average strike direction of the fractures of interest accurately enough, we have adjusted this sentence and others accordingly (e.g., line 549):

> "It can therefore be assumed that higher fracture permeability can be expected in the NNW-SSE direction, which can lead to the application of hydraulic stimulation."

**John Hooker:** *"497. Slickensides a filling material?"*

**Authors' reply:** You are right, slickensides are no filling material. We corrected the sentence accordingly (line 564):

> "More than half of all observations showed paleo filling materials like calcite beside calcite enriched slickensides on the fracture surface, approx. 58 % and 17 %, respectively."

**John Hooker:** *"534-5: P-waves more sensitive to pore spaces and unfilled cracks? I thought it was precisely because S-waves only travel through the solid rock that S-waves are more sensitive."*

**Authors' reply:** In fact, the sensitivity of P- and S-waves to unfilled pore space and crack volume cannot easily be generalized, since many factors (e.g. the crack geometry) have to be considered. However, if we were to consider a change in the saturation state, P-waves would provide more insight into the state of the rock than S-waves. Since the sentence mentioned is not important for the core statement of the paragraph, the sentence is deleted from the manuscript.

**John Hooker:** *"589. "comparable" any two numbers are comparable; you mean similar? How similar? Can you quantify intersection probabilities, if I drilled a hole at Dortmund?"*

**Authors' reply:** Indeed, the intersection probabilities could not be clearly quantified from the original manuscript submitted. Therefore, we have now added a simple method for estimating 2D fracture connectivity according to Ozkaya (2011) to the manuscript. This method allows to estimate the average number of discontinuity intersections per discontinuity in a 2D plane parallel to the exposed outcrop wall. We have added this paragraph to the methodology section (line 235):

> "In addition, the simplified 2D fracture connectivity of fractures aligned parallel to the examined outcrop face, that is, the average number of discontinuity intersections per discontinuity $P$, was estimated for each opencast mine according to Ozkaya et al. (2011)

> $$P = \sum_{k}^{n} d_k^{-1} \sum_{k}^{n} d_k \sum_{j \neq k}^{n} d_j L_j \sin (\Delta\theta_{jk}), \tag{3}$$

where $\Delta\theta_{jk}$, $d_j$, $L_j$ denote the angle between the average striking directions of two discontinuity sets $j$ and $k$, the number of discontinuities per total scanline length (i.e., fracture density), and the mean discontinuity length of each discontinuity set, respectively. In each outcrop we encountered three sets of fractures (n = 3) in the respective investigated carbonate layers. Therefore, the average fracture connectivity $P$ of each exposed carbonate layer was estimated on the basis of three average fracture densities and mean discontinuity lengths."

The results of the evaluation of the Wuppertal discontinuity data set are described starting at line 326:
"The 461 recorded discontinuities can be divided into three sets according to their orientations (Fig. 4b), whereby set 1, 2, and 3 could be assigned 181, 142, and 138 discontinuities, respectively. On average 2.71, 2.13, and 2.07 discontinuities or fractures were recorded per scanline meter in the sets 1, 2, and 3, respectively, corresponding to the average fracture density of the corresponding sets. For the sets 1, 2 and 3 the average discontinuity lengths amount to 7.01, 4.07 and 11.43 m, respectively. The angles between the average striking direction of set 1 and 2, set 1 and 3, and set 2 and 3 were derived as 112°, 62°and 50°, respectively. This results in an average number of discontinuity intersections per discontinuity of 21.42, that is, the estimated 2D fracture connectivity according to Ozkaya et al. (2011). The identified fracture interconnection in this outcrop is about 6 to 8 times higher than in the other outcrops Hönnetal and Hagen Hohenlimburg."

The results of the discontinuity data set of Hagen Hohenlimburg are described as from line 349:
The discontinuities recorded in this outcrop show three different strike directions (Fig. 5b) and were therefore divided into three sets with 134, 95, and 132 discontinuities. The fracture densities of set 1 to 3 were calculated as 1.76, 1.25, and 1.24 $m^{-1}$, respectively. The mean discontinuity length of the discontinuity set 1, 2, and 3 amounts to 1.24, 0.96, and 1.46 m, respectively. The discontinuity lengths determined here are the smallest of all recorded discontinuity sets of all quarries. The average strike difference direction between the sets $\theta_{1,2}$, $\theta_{2,3}$, and $\theta_{1,3}$ were calculate as 122°, 63°, and 59°, respectively. The value of fracture connectivity P determined for Hagen Hohenlimburg is 2.49 and corresponds to the smallest value of all examined quarries.

The results of the 2D fracture connectivity evaluation for the data from Hönnetal are given from line 368:
"In Hönnetal, too, the over 200 recorded discontinuities occur in three clusters or sets of striking directions. The sets 1 to 3 could be assigned 140, 24, and 82 discontinuities, respectively. The mean discontinuity length of the discontinuity sets $L_1$, $L_2$, and $L_3$ are 4.51, 5.57, and 2.85 m, respectively. The fracture densities $d_1$, $d_2$, and $d_3$ were derived as 2.23, 0.38, and 1.31 $m^{-1}$, respectively. Although the fracture density of set 2 is very low compared to the other sets, the mean discontinuity length is relatively high and reaches 5.57 m. The angles between the strike directions of sets 1 and 2, 1 and 3, and 2 and 3 are 115°, 87°, and 28°. The identified fracture properties of the sets result in an average number of discontinuity intersections per discontinuity of 3.71 for the exposed carbonate layer in Hagen Hohenlimburg.

In discussion chapter 4.2.3 the data are discussed and it is also apparent from these data that the reservoir exposed in Wuppertal is the most interesting carbonate reservoir for a deep geothermal project (line 593):
"However, our results of the 2D fracture connectivity analysis clearly show that the reservoir in Wuppertal has the highest average 2D fractureconnectivity. Both the fracture density and the mean discontinuity length of each discontinuity set are significantly higher in Wuppertal

than in the other outcrops. Accordingly, we can assume that this reservoir probably has considerable geothermalpotential. For the Hönnetal reservoir a low number of discontinuity intersections per discontinuity was determined. One reason for this relatively low value is the location of the reservoir on the anticline axis and the associated tectonics. Although the recorded fractures in Hagen Hohenlimburg exhibit the greatest variety of strike directions, the estimated 2D fracture connectivity is thelowest at this location, which is due to the short discontinuity lengths and the associated low fracture densities. Nevertheless, based on the observations of many karst formations and altered host rocks by hydrothermal veins in Hagen Hohenlimburg, we conclude that there might be a higher permeability in the Hagen Hohenlimburg reservoir and in the surrounding areas than in the other studied areas. But, the distribution of this reservoir in the subsurface is still under debatte (e.g., Salamon and Königshof, 2010) and its conditions due to karstification still have to be proven."

In conclusion, we now state that in the target horizon, which is located below the major cities of Essen, Bochum, and Dormund, we expect similar fracture connectivity as in the investigated quarries. This expectation is based on the combined evaluation of the complete data set of the manuscript. The corresponding sentence in the conclusions reads (line 670):
"Besides, taking all measurements into account, we expect similar probabilities for interconnecting fracture networks in the deep reservoir. From this it follows that the recorded field surveys (Wuppertal, Hagen Hohenlimburg, and Hönnetal) can be assigned representatively for the reservoir of interest below the cities Essen, Bochum, and Dortmund."

---

## Author Comment (AC2) · 11 Aug 2020

**Reviewer 2 (Sadegh Karimpouli)**

**Sadegh Karimpouli:** *"Geological map lacks of enough data. I mean, all information in the text must be transferred into the geological map. Then, in a regional point of view, all data could be connected, judged and concluded. The most important points are:*
    ***i.*** *Putting rose diagrams of all sites on the geological map*
    ***ii.*** *Showing strike and dip of the target layers on each site and if possible on the other areas.*
    ***iii.*** *Showing orientation of the dominant and/or present day maximum stress (either from literature or the authors observation)."*

**Author's reply:** Thank you very much. You have listed a very good improvement point here, which we were very pleased to implement. Based on your comment, we have added the points mentioned above to the original Figure 1. The cross-sections have now been merged into a separate figure (Fig. 2).

**Sadegh Karimpouli:** *"I tried it as the attached figure. Based on maps in my figure, Hönnetal site is located on an anticline axis. The authors should be careful about combining fractures orientation data from this site with other sites. They should explain:*
    ***a.*** *How are those rose diagrams are connected to each other?*
    ***b.*** *How are they connected to the regional tectonic regime?*
    ***c.*** *Line 417: "The foundation of our model is an approx. 300 m thick carbonate layer, dipping northwards at a shallow dip angle of about 30 to 40◦". Fracture orientations on the anticline axis (HLO) show a different pattern compared to downward limbs (WHO, HKW). How do the authors combine them together?"*

**Author's reply:** We thank you for pointing out that our fracture results from the individual quarries are not sufficiently explained and discussed with regard to (a) common features, (b) the tectonic regime and (c) the formation of the regional Remscheid-Altena anticline. Your questions and similar comments by the other reviewer (John Hooker) have prompted us to revise the text to adequately address and answer the questions raised (line 469):

> „All studied outcrops are located in the large scaled fold formation Remscheid-Altena anticline. However, there is a disagreement between the three outcrop results which might be an effect of the formation of the regional Remscheid-Altena anticline, different stress states, or different time of origin (Table 3). Due to the anticline formation, the strike directions of the present fractures in this region exhibit a rotation from the northern limb (Wuppertal) towards the tip of the anticline (Hönnetal). Fractures striking NE−SW are highly related to folding mechanism and are parallel oriented to fold axes which have been studied within the Rhine-Ruhr area (Drozdzewski, 1985; *Brix et al.*, 1988; DEKORP Research Group, 1990; Drozdzewski and Wrede, 1994). The dominant fracture strike directions NNW−SSE in Wuppertal agree with the structure of the regional Remscheid-Altena anticline (Fig. 1b) and the overall assumed mean principle stress direction according to the World Stress Map (Heidbach *et al.*, 2016) and additional available stress data (Rummel and Weber, 1993). In western Germany, or to be more precise in North Rhine-Westphalia, the World Stress Map contains a wide variability of mean principle stress directions (Heidbach *et al.*, 2016), that can be explained by shallow stress measurements, local anomalies which can be attributed to weak coal-seems, or regional NE−SW thrusts. The observed strong scattering of the fracture strike directions in the dolomitic carbonates exposed in Hagen Hohenlimburg is due to their formation by hydrothermal veins during the Hercynian Orogney (Gillhaus *et al.*, 2003). Furthermore, Gillhaus *et al.* (2003) explain that the existing NNW−SSE striking fractures are of post-Hercynian Orogeny origin. The cause of the slightly different fault strike directions in Hönnetal cannot be clearly specified according to the current state of scientific knowledge.

> Most likely, the fracture formation can be explained by various local and temporal stress anomalies and different formation times."

**Sadegh Karimpouli:** *"Section 4.2.1: The authors explain "The main discontinuity orientations were documented as NNW-SSE, NW-SE, and NE-SW." and then conclude "we propose to focus on discontinuities that are approximately oriented N-S for future shallow geothermal applications.". Is N-S one of the main directions or what? Is this conclusion on the basis of maximum stress direction? How is the contribution of the other factors such as fracture filling, conductivity and so on?"*

**Author's reply:** After re-examining the text passages, we fully agree that in Section 4.2.1 not all results have been taken into account to adequately explain which fracture orientation is of interest. To specify the direction more precisely, we have concluded to rename N-S-striking fractures to NNW-SSE-striking ones, which corresponds to our results and is also comparable to the assumed mean principal stress direction (line 545):

> "When comparing the recorded discontinuity orientations with the orientation of the maximum horizontalstress (Heidbach *et al.*, 2016), we propose to focus on discontinuities that are oriented NNW-SSE for future geothermal applications, which are highly probable filled by calcite."

We would also like to thank you for pointing out that we need to look at the different filling material to substantiate our assumption. The new Figure 7 shows rose diagrams of all measured discontinuity sets as a function of their fracture filling, that is, whether the fractures are filled with (a) calcite or (b) debris. We have reformulated section 4.2.1 in order to link the individual results more clearly and to justify the discontinuity direction of interest (line 540):

> "In addition, we present fracture orientations versus filling materials, these are, calcite or debris. The orientation of the recorded veins allows us to conclude, that many of the discontinuities studied on the outcrop scale can be related to residual stress and stress release during unloading regimes (cf. Nickelsen and Hough, 1967; Roberts, 1974, Fig. 7)."

Furthermore, in line 549 we go into more detail about the fracture fillings and the resulting implications:

> "It can therefore be assumed that higher fracture permeability can be expected in the NNW–SSE direction, which could be of interest for the application of hydraulic stimulation techniques. In addition to the relative orientation of the fractures to the direction of the main principal stress, the filling is also decisive for whether the fractures are potentially open or closed in the subsurface (Laubach *et al.*, 2004). Thus, there might be open fractures that are not necessarily aligned in the direction of the principal stress and are still open. This is particularly true for fractures that are filled, for example, with cement (Laubach *et al.*, 2004)"

**Sadegh Karimpouli:** "Use different parameter for thermal connectivity and discontinuity frequency (both of them are $\lambda$)."

**Author's reply:** Thank you very much for the hint. In the manuscript, the symbols for thermal conductivity, $\lambda_{dry}$, and discontinuity frequency, $\lambda$, have already differed, but for a clearer distinction, we will gladly accept your suggestion. Both in the text and in the tables the symbol $\kappa_{dry}$ is now used for thermal conductivity.

**Sadegh Karimpouli:** "Figure 6: How do you translate connected pores more than total pores?"

**Author's reply:** Your observation is very correct. Physically, it is possible that the connected porosity exceeds the total one. Taking into account the measurement inaccuracies that may be present in the experimental determination of these volume properties, however, this is possible. The connected and the total porosity are determined with different methods and different errors. Therefore, we have already pointed out these errors in the text (line 340:)

"Taking into account the measurement uncertainties, the total and connected porosities overlapped (Fig. 8a).").

---

## Author Response (AR2)

**Topical Editor (Randolph Williams)**

**Randolph Williams:** *"There are currently two problems with the current version of the manuscript that prevent its publication. The first concerns the quality of the presentation / writing. There are some sections of the text where far more detail than necessary is provided. Other sections of importance do not include nearly enough. As an example of the latter, important corrections and quantitative methodologies simply refer to a previous publication. You cannot of course fully redescribe methodologies produced by others here, but at least a general description of their function and purpose is required for the reader to grasp your work."*

**Author's reply:** First of all, thanks for the extensive review, comments and suggestions. We have tried to implement your review as good as possible. Examined from temporal distance, we were able to understand many points of criticism. However, it is difficult for us to understand your criticism of the scope and detail of the methodology. We have described the 1D scanline sampling technique, the estimation of the 2D fracture connectivity as well as the different laboratory investigations in an extraordinary scope. For example, in comparison to all other publications that use the scanline method of Priest and Hudson (1981), our description has an extraordinary scope and therefore only our description enables the reader to apply the methodology independently.

**Randolph Williams:** *"In addition, the discussion often refers to the results section in only a qualitative manner (i.e. this location had higher 2D fracture connectivity than another location). Ideally, these metrics are all included in a figure or table that the reader can be referred to when making these and similar statements. Alternatively, the actual values can be reiterated in the discussion for clarity. In the current version, neither is done, which requires the reader to either remember the values or flip back through the paper. Both are troublesome given that the text is really quite verbose and long."*

**Author's reply:** Based on your comments, we have rewritten and greatly shortened (from 2 pages to 1 page) the section "Field Discontinuity Observations". In addition, we have now created clearer cross-references to the figures and tables we have created. As requested, the section now gives a general overview of the relevant findings. Furthermore, we have adapted the discussion accordingly and included clearer cross-references. Some values are also reintroduced in the discussion, so that the reader can follow the train of thought of the discussion better.

**Randolph Williams:** *"There are also many grammatical mistakes with respect to english language grammar. Admittedly, most of these mistakes are minor, but there is a sufficient amount of them that makes reading difficult at times. I have noted many of these instances below, but the list is not exhaustive. I understand that Solid Earth is an international journal, and many of the authors do not have english as their native language. This is not inherently a problem, but I would ask that you attempt to address some of these issues in your next version."*

**Author's reply:** We have taken another look at the grammar and language of the complete manuscript and improved both points of criticism in the current version of the manuscript.

**Randolph Williams:** *"The second issue concerns the length of the manuscript with respect to its "scope" or conclusions. I would like to see this manuscript be greatly shortened and streamlined. As it currently stands, the length is far above what is reasonable for the amount of data, their complexity, and what can be confidently concluded. Too much extraneous detail is included. For example, the geologic setting section is very detailed (although a bit difficult to follow), but very few of these details are referred back to in framing the results and subsequent discussion. This suggests that the level of detail included is more than what is*

*required. The fracture fill and roughness data are similarly interesting, but do not seem to play a major role in your overall conclusions. Veining in particular is tough to extrapolate, because assuming that the veins do not predate the fold, then there is a difference in burial between the observed outcrops and the subsurface reservoir of 4 to 6 km. This would result in huge differences in local geochemical conditions, mineral kinetics, etc."*

**Author's reply:** Thank you very much for this remark. The manuscript has been shortened and streamlined. Especially the sections "Geological Setting" and "Study Areas" were shortened considerably. Furthermore we have adapted the "Results" and the "Discussion" so that the reader can follow the discussion more clearly. We have also rewritten text passages to discuss the filling material and roughness.

**Randolph Williams:** *"The relevant story, as I see it from your data and conclusions, is: 1) the matrix permeability of the unit that is extrapolated into the subsurface is insufficient for energy production; 2) the unit at depth is likely fractured, where fractures parallel to the max horizontal stress may still be open; 3) the length, density, and orientation of fractures at the surface suggests good connectivity at depth, assuming the network is similar there; 4) Thus, the unit has potential for geothermal energy production. In that way, the study is useful and therefore deserving of eventual publication. Assuming that I understand these central conclusions correctly, the manuscript should be greatly streamlined around these ideas to emphasize them. My estimate is that the length of this manuscript could reasonably decrease by as much as 30% without sacrificing detail or clarity. In fact, I think the clarity would be greatly improved as a result."*

**Author's reply:** The manuscript was shortened and the focus was reduced only to the essential conclusions.

**Line-by-line comments:**

**(A) Abstract**

**Lines 4 to 5:** Please revise this sentence for grammar/clarity. Please pluralize "depth". What is a permeability "rate"? How does one study "permeability rates with sufficient conductivity"?
*Author's reply:* Pluralized "depth" in line 4. Sentence changed accordingly (line 4 to 5): *"Matrix permeability and thermal conductivity was mainly studied in karstified carbonates from the Late Jurassic reef facies."*.

**(B) Introduction**

**Introduction:** This is generally well written from a language clarity perspective. It is also, however, a bit less focused than would be ideal. I think the paper would be very well served by a judicious rewrite of the introduction, with a specific emphasis on getting to the "point" of this study a bit more quickly and with only the relevant information included. At the very least, some additional figures (e.g. maps) would be very helpful in making the first paragraph accessible to the reader.
*Author's reply:* We have shortened and restructured the introduction to focus more clearly on the characterization of fractures in potential reservoir rocks for geothermal applications in the Rhine-Ruhr metropolitan area, Germany, which is also the title of the manuscript. The Rhine-Ruhr area is an important study area for renewable energy, especially geothermal energy, due to its economic history and the resulting infrastructure, population density and geological setting.

**Line 51:** Please change "associate" to "associated".
*Author's reply:* The introduction was restructured and shortened, and in the process the mentioned expression or the whole sentence was removed from the manuscript.

**Line 55:** What exactly is meant by "mechanical conditions"?
*Author's reply:* The introduction was restructured and shortened, and in the process the mentioned expression or the whole sentence was removed from the manuscript.

**Line 57:** Please change to "...geological ambiguity and to characterize...".
*Author's reply:* The introduction was restructured and shortened, and in the process the mentioned expression or the whole sentence was removed from the manuscript.

**Line 63:** What exactly is meant by "stability" in this case?
*Author's reply:* We were referring to the stability (i.e., strength) of the reservoir. We have adjusted the sentence. The corrected sentence reads (line 47 to 48): *"Fractures and fracture networks have a decisive influence on the fluid flow within the reservoir (e.g., Odling et al., 1999), but also on its stability (e.g., Cappa et al., 2005)."*.

**Line 73:** Please change to "...at outcrop and sample scales".
*Author's reply:* We have revised the sentence in line 59.

**Line 74:** Please change "fractured" to "fracture".
*Author's reply:* We have changed the expression in line 60.

**Line 77:** I do not understand this line as written.

**Author's reply:** *We have revised the sentence and it reads (line 62 to 63): "All quaries are located within the Devonian Reef Complex. The dominant stratigraphic unit in thosequarries is the Massenkalk Fazies from Middle and Upper Devonian (Krebs, 1970; Paeckelmann, 1979; Schudack, 1993).".*

**Lines 82 to 84:** I am not sure this foreshadowing is necessary, but ultimately it is your choice.

**Author's reply:** *We think that the foreshadowing is necessary at this point.*

**(C) Geology, Outcrop Investigations, and Laboratory Measurements - Geological Setting**

**Geological Setting:** As stated above, I think this paper would benefit from some streamlining. Part of that would involve considering exactly which aspects of the geologic history are strictly relevant. If kept more or less as is, substantial revising for english-language clarity will be required.

**Author's reply:** *We have decided to keep the geological setting within the scope, but we have reviewed and revised the language and grammar. The manuscript is the first in a series of publications dealing with geothermal energy use in the Rhine-Ruhr area. As the first manuscript in this series, this manuscript is intended to serve as a pilot study and introduction and therefore all relevant geological information on the carbonates in the Rhine-Ruhr region will be collected and listed.*

**Line 87:** Please revise for language clarity.

**Author's reply:** *We have revised the sentence and it reads (line 72 to 74): "The Rhenohercynian Massif is the northernmost mountain belt of Europe, which results from a shallow marginal sea surrounded by Laurussia (Old Red Continent and Baltica) in the north and the microcontinent Avalonia in the south (Kossmat, 1927).".*

**Line 90:** Revise for clarity/grammar.

**Author's reply:** *We have revised the sentence and it reads (line 75): „A shelf sea formed above the thin crust, which is called the Rhenish-Hercynian Basin.".*

**Lines 92 to 93:** Revise for clarity/grammar.

**Author's reply:** *We have revised the sentence and it reads (line 77 to 78): "During the Hercynian Orogeny, a tectonic NW-ward movement led to expansion as well as periodic marine transgressions and regressions.".*

**(D) Geology, Outcrop Investigations, and Laboratory Measurements - Study Areas**

**Study Areas:** I feel there are many redundancies between this section and the previous. Ultimately, they could be easily combined and substantially reduced in length.

**Author's reply:** *In the course of the overall shortening, we have streamlined the section "Study Areas" and deleted possible repetitions with the section "Geological Setting".*

**Line 135:** How is a tectonic setting "massively affected" by an anticline? This seems to be putting the cart before the horse.

**Author's reply:** *We have revised the sentence and it reads (line 114 to 115): "The regional tectonic setting has resulted in the Remscheider Sattel, the Ennepe-Thrust, and the Großholthausener fault, in Hagen-Hohenlimburg (Fig. 1a).".*

**Lines 171 to 174:** I am having a great deal of trouble understanding these lines as written. Also, I am not sure I would describe beds with a dip direction of ~330 as comparable to those with a dip direction of ~020. That is nearly 50 degrees of difference.

**Author's reply:** Thank you for pointing out that the wording of this paragraph is not understandable. We agree with you that a deviation of about 50° should not be called "comparable". We have revised this paragraph and it reads (line 136 to 138): "*The mean bedding orientation in the quarries Wuppertal, Hagen Hohemlimburg and Hönnetal is approximately 327°, 345°, and 18° in dip direction with an average dip angle of about 37°, 42°, and 28°. Each stone pit's working levels were approximately oriented perpendicular to each other and strike NNW–SSE and NE–SW (Fig. 3).*".

**Lines 176 to 179:** I think you could reasonably just call them "fractures", specifying that the term includes veins but does not include open space along bedding planes or schistosity. It is in the title of the paper afterall. I appreciate the attempts to be explicit, but this is a journal for geoscientists.

**Author's reply:** Thank you for the comment. However, we think a clear classification is crucial what kind of fractures we have studied and which not. Therefore, we have not changed the sentence.

**Lines 193 to 194:** Please revise for clarity.

**Author's reply:** We have revised the sentence and it reads (line 158 to 159): "*The tape's line levelling is defined by the angle of plunge (approx. $0° \leq \theta_S \leq 15°$).*".

**Line 203:** Again, this is a journal for geoscientists. You could very easily just say "The dip and dip direction of the intersecting fractures were recorded".

**Author's reply:** We have revised the sentence and it reads (line 168): "*Orientation: The dip and dip direction of the intersecting fractures were recorded.*".

**(E) Results - Field Discontinuity Observations**

**Field Discontinuity Observations:** In light of the provided tables, this exhaustive description of the data really does not seem necessary. A general overview of the relevant findings would suffice.

**Author's reply:** Based on your comments, we have rewritten and greatly shortened (from 2 pages to 1 page) the section "Field Discontinuity Observations". In addition, we have now created clearer cross-references to the figures and tables. As requested, the section now gives a general overview of the relevant findings.

**Lines 305 to 310:** Suggest using azimuth ranges for fracture sets, as they are far easier to understand than long strings of NNW-SSE, NE-SW, etc.

**Author's reply:** We have revised the sentences and they read (line 275 to 279): "*In Wuppertal the sets were grouped intothe azimuth directions 172° (181 discontinuities), 55° (142 discontinuities), and 105° (138 discontinuities) (Fig. 4c,d; Table6). In contrast, the discontinuity sets in Hagen Hohenlimburg and Hönnetal were oriented towards 135° (134 discontinuities), 12° (95 discontinuities), and 75° (132 discontinuities), and 176° (140 discontinuities), 65° (24 discontinuities), and 87° (82 discontinuities), respectively (Figs. 5c,d; 6c,d; Table 6).*".

**Lines 313:** "Paleofilled"?

**Author's reply:** We have changed the expression to "calcite-filled".

**Line 319:** What is the differentiation between a filled fracture and a closed fracture?

**Author's reply:** Thank you for pointing out that we have not yet clearly clarified this distinction. We have added the following sentences to Table 1: "Overview of the documented discontinuity properties and their classification as defined for scanline surveys (modified after Markovaara-Koivisto and Laine, 2012). In our observations we distinguish between open, closed, and filled fractures. Open fractures arethose that are neither filled with calcite nor with debris. Closed fractures are in fact joints. Filled fractures are not categorized as closed ones,since both fracture surfaces are not in direct contact with each other. Filled fractures can either be filled with calcite or with debris.".

The original sentence was deleted due to shortening.

**Line 320:** Does the use of the term significant imply that a statistical analysis was applied?
**Author's reply:** We have now supplemented the manuscript with a statistical analysis of the strike directions with respect to their filling material and no significant difference between their strike directions can be found (Fig. 7). Accordingly, we have applied this insight to the entire manuscript and adapted the corresponding text passages.

The following statement can now be found in the results section (line 280 to 282): "*By further differentiating the strike directions of the classified discontinuity sets further according to their filling material no strike rotation between calcite and debris-filledfractures were identified (Fig. 7).*".

In the discussion it reads now for example: "*We could not detect any rotation between the strike direction of calcite and debris-filled fractures (Fig. 7). Hence, the maximum horizontal stress may have changed only slightly, if at all, in the time between the formation of the calcite and the debris-filled fractures.*".

**(F) Results - Laboratory Characterizations**

**Lines 397 to 409:** No tables or figures to cite with this information?
**Author's reply:** We have adjusted the sentences and added cross-references to Table 4 (line 310).

**Lines 410 to 411:** Sorry, but I am unable to follow this sentence?
**Author's reply:** We have revised the sentence and it reads (line 321 to 322): "*The thermal conductivity measurements showed similar results for limestones as for dolomitic carbonates. However, a slight scattering can be observed.*"

**Lines 423 to 448:** So you also did all these analyses on a single vein sample? Can you provide some text justification for why? What was the goal?
**Author's reply:** Due to the suggestion to shorten the manuscript and thus focus on the main part of the study, we have decided to eliminate the paragraph. We agree with the comment that these findings do not add significant value in the present study. Overall, we have deleted all information, descriptions, etc. that refer to this particular sample from the manuscript.

**(G) Discussion - Estimating the Geological Subsurface Model of the Carbonate Reservoir**

**Lines 470 to 485:** The reader cannot reasonably assess your interpretations regarding the relative geometry of the fractures and regional structures unless you reference some stereonets that also include the orientation of the regional structures. Reviewer #1 suggested that the fracture orientations be restored to

their "original" orientations through an unfolding procedure. This is a good suggestion, and it is not clear to me why that was not done? I do not agree that this "exceeds" the scope of the study as stated in the response letter. It would seem to be well within the scope of the study.

Author's reply: Many thanks for the explicit advice to deal with the rotation of the measured fractures in detail once again. In the scope of the revisions we have applied a simplified method to all our measurements. Now you can see the recent measured fracture orientations from the stereograms and the rotated fracture orientations from the corresponding rose diagrams (Fig. 4a, 5a, 6a and Fig. 4b, 5b, 6b). We have adapted and changed the text accordingly (line 364 to 368): "*All studied outcrops are located in the large-scaled Remscheid-Altena anticline formation. However, there is a difference between the three outcrop results, which could be influenced by the regional the Remscheid-Altena anticline, different stress conditions and/or a different time of origin (Table 3). The fracture orientations were rotated according to the corresponding bedding layers (Figs. 4a, 5a, 6a) to a horizontal position (Figs. 4b, 5b, 6b). The fracture unfolding procedure indicates a similar tectonic origin for fractures sets from Wuppertal and Hagen Hohenlimburg.*".

**(H) Discussion - Orientation of the Fracture Network**

Lines 541 to 543: Please be more specific as to why this interpretation is valid. Only a citation is given. Evidence should also be provided.

Author's reply: We agree, that the sentence as written needed an evidence to prove the statement. However, we cannot prove this interpretation with sufficient accuarcy. Therefore, we have changed the sentences as follows (line 437 to 441):"*Nickelsen and Hough (1967) and Roberts (1974) investigated discontinuity orientations in sedimentary layers with respect to the maximum horizontal stress and their corresponding filling material. The orientations of the respective structures in their area are almost identical to the values we measured. They concluded that the discontinuity directions can be related to residual stresses and stress release during past unloading regimes, which might be transferable to our studied areas (Fig. 7).*".

Lines 546 to 547: So, you feel that this fracture set is most promising for exploiting geothermal energy, but you also think it is "highly probably" filled with calcite cements? I`m confused. Are the calcite cements only occluding a small portion of the fracture porosity? Else, sealed fractures generally make for poor conduits, no?

Author's reply: You are absolutely right in claiming that filled fractures are bad conduits. We could not find any difference in orientation or occurrence between calcite and debris filled fractures. Therefore, the filling material cannot be the only criterion we have to consider. A second criterion must therefore necessarily be the direction in which we prefer to find open rather than closed cracks, relative to the maximum horizontal stress in the region. Fractures oriented in the direction of the maximum horizontal stress are very promising for enhanced fluid flow. Even if we cannot predict it with certainty, there is a chance that these fractures are open due to the orientation towards the maximum horizontal stress. In addition, the hydraulic stimulation of these fractures and/or fracture networks could be very promising and allow the operation of geothermal energy systems.

This information is now included in the following sentences of the discussion (line 455 to 464): "*More than half of all our observations showed paleo filling materials such as calcite and in addition, approximately 17 % showed calcite enriched slickensides on the fracture surface (Fig. 7). Thus, apart from the orientation, no differences in occurrence between calcite and debris-filled fractures can be detected, and we must further*

*consider the orientation of the fractures. Even if we cannot predict it with certainty, pre-existing discontinuities oriented parallel to the principal horizontal stress could be observed openly, while discontinuities oriented perpendicular to this stress tend to be closed (Lorenz et al., 1991). Accordingly, the highest fracture permeability can be expected in NNW–SSE direction. But there might be open fractures that are not necessarily aligned in the principal stress direction and are still open. This is particularly true for fractures that are filled, for example, with cement (Laubach et al., 2004). In general terms, a large number of fractures might be paleo fluid-flow paths striking in NNW–SSE direction, which could be exploited by advanced drilling methods (e.g., hydraulic fracturing; Dahi Taleghani and Olson, 2013) for future geothermal applications."*.

**(I) Discussion - Filling and Surface Roughness of the Fracture Network**

**Section 4.2.2:** This is really quite difficult to follow.
**Author's reply:** We have now renamed the section to "Orientation, Roughness, and Filling of the Fracture Network" and have completely revised and shortened it accordingly. As you mentioned before, we have now extended the discussion of the filling material.

**Lines 562 to 563:** What is the typical aperture of these fractures? Seems like the probability of detecting them seismically is extremely low? There is almost certainly information already out there that would allow you to evaluate this claim.
**Author's reply:** Thank you for the comment to mention already a possible solution for the detection of the mentioned fractures, which are of interest. We have adapted this idea in the following sentences (line 467 to 470): *"Considering the typical seismic resolution concerning the depth of investigation and fracture aperture, individual discontinuities will probably not be resolved (Martin et al., 2006). However, the fracture density may be sufficient to demonstrate macroscopic anisotropy using seismic methods (Löer et al., 2018)."*.

**Line 565:** What does the percentage of calcite filled fractures have to do with any specific tectonic origin?
**Author's reply:** Your observation is correct. This was a textual relic. The sentence was deleted.

**Line 566:** A "slight" rotation. Did you evaluate statistically if there is a significant difference between the two orientations. Lots of methods are available to do this. See paper by Roberts et al. JSG, 2019 on statistical analysis with structural data. Figure 7 seems to show a slight rotation only at Honnetal?? The other two cites seem essentially the same.
**Author's reply:** We have now supplemented the manuscript with a statistical analysis of the strike directions with respect to their filling material and no significant difference between their strike directions can be found (Fig. 7). Accordingly, we have applied this insight to the entire manuscript and adapted the corresponding text passages.

The following statement can now be found in the results section (line 280 to 282): *"By further differentiating the strike directions of the classified discontinuity sets further according to their filling material no strike rotation between calcite and debris-filledfractures were identified (Fig. 7).".*

In the discussion it reads now for example (line 450 to 452): *"We could not detect any rotation between the strike direction of calcite and debris-filled fractures (Fig. 7). Hence, the maximum horizontal stress may have changed only slightly, if at all, in the time between the formation of the calcite and the debris-filled fractures.".*

**(J)  Discussion - Connectivity of the Fracture Network**

**Line 580:** Field scale?

Author's reply: We have changed the sentence structure to avoid misunderstandings. The sentence now reads as follows in line 469: *"Predominantly discontinuities with short trace lengths occur on the field scale Figs. 4d, 5d, 6d)."*.

**Line 587:** Seems highly probably that they are underestimated rather than "might be".

Author's reply: We have revised the sentence and it reads (line 473): *"In other words: 40% of all measured trace lengths are highly probably underestimated by the method."*.

**Line 585:** That analysis needs to be summarized. This manuscript has too much of a tendency to just refer the reader to other sources on important points of methodology or conclusion. That is reasonable to do only in some cases. This is not one of them, as the reader cannot be expected to read so much external material in order to understand what was done here.

Author's reply: This methodology was already summarized in lines 194 to 198. The sentence was rewritten and the cross-reference to Equation 2 was inserted (line 478): *"This source of error was attempted to be minimized by the censored semi-trace length analysis (Equation 2)."*.

**Line 593:** I never saw a probability of interconnection listed, so it is not clear what "comparable" to that would be.

Author's reply: This sentence was deleted due to shortening of the section.

**Line 594:** Again, this statement seems very qualitative. You have quantitative estimates. Please refer back to them here explicitly (i.e. the actual values). Better still, refer to a figure or table that includes them.

Author's reply: We have adjusted the sentence according to the comment (line 482 to 483): *"The results of the 2D fracture connectivity analysis clearly show that the reservoir in Wuppertal has the highest average 2D fracture connectivity, that is, an average number of discontinuity intersections per discontinuity of 21.42 (Table 6)."*.

**Line 597:** Again, please reference the actual estimates. Moreover, why do these metrics not appear in any figures or tables?

Author's reply: We have adapted the sentence according to the commentary and added an additional table to the manuscript (line 487 to 490): *"Although the recorded fractures in Hagen Hohenlimburg exhibit the greatest variety of strike directions (Fig. 5), the estimated 2D fracture connectivity is the lowest at this location due to the short discontinuity lengths and the associated low fracture densities (Table 6)."*.

**Line 605:** Not overly clear to me what this paragraph has to do with fracture connectivity?

Author's reply: In this paragraph, we discuss pore connectivity, which can also influence permeability if the pore connectivity is high enough. To emphasize the correlation even more, the following sentence in line 492 has been adapted: *"Consequently, a higher density of potentially hydraulically stimulated pores and cracks (or fracture network) can be expected in the dolomitic carbonate layers on the field scale."*.

**Line 635:** This is one vein. How do we use it to infer anything about a target horizon?

**Author's reply:** This sentence was deleted due to shortening of the section.

**(K) Conclusion**

**Line 640:** So to summarize results: 1) matrix permeability would be insufficient for geothermal energy production; 2) Fractures may be present at depth based on outcrop, and those parallel to the max horizontal stress are most likely to be open and therefore productive; 3) Based on fracture length and orientation observed at the surface, there may be a well connected network at depth. These are useful conclusions, but they could be stated much more succinctly.

**Author's reply:** We have now restructured the conclusions. Our conclusions are now more accessible.

[revised manuscript text omitted]

---

## Author Response (AR3)

**Topical Editor (Randolph Williams)**

Dear Mr. Williams,

once again many thanks for the helpful comments and remarks. You have raised good points and we have worked on improving these last points as well. We also followed up on your comments and re-read the whole manuscript twice and made improvements where necessary. Based on your comments and your stimulating questions, as well as the reviewers', the quality of the manuscript could be further improved. Thank you!

On behalf of all co-authors,

Yours sincerely,
Martin Balcewicz

**Randolph Williams (general comment):** "*Joints versus discontinuities. Used interchangeably. Either stick with discontinuity throughout, being clear about which type in discussion when relevant, or discard discontinuity and simply reference the structure type (e.g. joint vs shear fracture vs vein) directly in each relevant instance.*"
Author's reply: Thank you very much for this comment. We have chosen to use the term discontinuity throughout the manuscript, except in those parts where we explicitly mean fractures, joints or veins.

**Line-by-line comments:**

**Line 6:** "…FOR which…"
Author's reply: We have corrected the sentence.

**Line 10:** "*Discontinuities" is too vague for the abstract. Please state exactly what was measured / observed in a paranthetical.*"
Author's reply: You are right that the expression is too vague. We have now indicated exactly what kind of discontinuities were measured. The sentence reads as follows (lines 10 to 12): "*During field surveys, 1068 discontinuities (139 open fractures without any filling, 213 joints, 413 veins filled with calcite, and 303 fractures filled with debris deposits) at various spatial scales were observed by scanline surveys.*".

**Line 21:** Sorry, I do not follow the logic here of why filled fractures would be a particular aspect that one would want in a geothermal system. Hopefully that is explained later in the manuscript. No rationale is presented here as to why those fractures would necessarily be re-opened or experience any dissolution.
Author's reply: Yes, you are of course right that in a geothermal system no calcite-filled veins are preferred. This statement has been very unfortunate so far and goes back to the original version, in which we concentrated too much on these veins and the associated hydrochemical solutions. The argument, which has now been formulated much more clearly, is that we assume that the majority of the fractures at depth should be material-free. We have removed the subject of hydrochemical solutions to avoid giving a false impression of the preferred fractures. The sentence now reads as follows (lines 21 to 23):
"*… and because only about 38 % of the fractures were observed with a calcite filling. The remaining fractures either showed no filling material or showed debris deposits, which we interpret as open at depth.*".

**Line 25:** I think this last sentence is unnecessary. Moreover, I do not understand its link with the rest of the abstract. The fracture reactivation discussion here is presented without any background.

**Author's reply:** Based on the implementation of your previous comment, we have deleted the sentence noted here from the manuscript.

**Lines 28 to 44:** On some level I appreciate this background, but it really is not necessary. The introduction would be well served by being yet more focused (although I do appreciate your previous efforts to provide more focus relative to the original submission). Honestly, if you simply used the text in lines 46-69 as the introduction that would be sufficient, but adding in a few of the details from the preceding paragraphs would also be helpful. This is more a style issue than anything else so the choice is yours, but that is my recommendation.

**Author's reply:** Many thanks for this suggestion, which we really appreciate. You must know that the Rhine-Ruhr area is really in an acute turnaround right now and many projects are planned in the field of geothermal energy. We would like to shorten the introduction, but the development of the Rhine-Ruhr area is an important motivator for the manuscript, in addition to the scientific stimulus. As already mentioned too often, our study is a kind of feasibility study and also the first of a long series of upcoming publications. Accordingly, we would like to emphasize here the importance of the Ruhr area in the first publication. In keeping with this, we would like to leave the introduction unchanged, although we understand your point very well.

**Line 55:** Remove capitalization of "Since".

**Author's reply:** We have corrected the sentence.

**Line 63:** Please adopt the English spelling of "facies" for this manuscript.

**Author's reply:** Many thanks also for this hint. We have changed the printout accordingly and incorporated this change into the entire manuscript.

**Lines 71 to 72:** How does a mountain belt "result" from a shallow sea? Please revise for clarity.

**Author's reply:** Yes, you're absolutely right. Thanks for the hint. The sentence made no sense in its conventional form. In order to reflect this thought sensibly, however, we would have to provide the reader with much more information. However, in view of the all-encompassing shortening, we have decided to only refer to the tectonic characteristics of the massif. These are directly related to our measurements and provide the reader with essentialbackground knowledge to evaluate the results in a context. The sentence now reads as follows (lines 73 to 74): *''The Rhenohercynian Massif is the northernmost mountain belt of Europe and shows the results of the Hercynian Orogeny with its numerous northeast striking folds and thrusts at the northern edge of the Variscian mountains.''*.

**Line 77:** Expansion of what? The sea? How is that differentiated from the idea of transgression which is subsequently mentioned? Eustatic sea rise vs subsidence related?

**Author's reply:** After another reading, we noticed significant mistakes, which are primarily due to the unsuccessful shortening. For this reason, we have revised the above sentence and others to reflect the geological setting correctly. We have limited ourselves to important findings only, which should lead to an improved reading flow. The idea that was established in the original sentence can now be found in lines 83 to 85: *''In the Late Devonian, the first coral reefs disappeared in the course of the Late Devonian extinction (Kellwasser event), which was first favored by a marine transgression and then by rapid regression (e.g., Meschede, 2018). ''*

**Line 84:** Crustal expansion? What exactly is meant by this?

**Author's reply:** This sentence was completely removed in the course of the mentioned revisions due to the comment to Line 77.

**Line 114:** How does a regional tectonic setting result in an anticline or any other specific structure?

**Author's reply:** Thank you for drawing our attention to linguistic ambiguities. We have changed the sentence as follows (lines 116 to 117): *''The regional tectonic structure is characterized by the Remscheider anticline, the Ennepe-Thrust, and the Großholthausenerfault, in Hagen-Hohenlimburg (Fig. 1a).''.*

**Line 174:** If the fractures have any significant roughness at outcrop scale then the aperture by definition is variable. Can you elaborate on this measurement? Was it a minimum, maximum, or else somehow "representative"?

**Author's reply:** Here you address a good point, which we have gladly revised. The determination of roughness and apertures were estimated by geometrical (qualitative) analysis. Accordingly, these are subjective data, but they were always determined by the same person. Therefore, these values depend on the personal assessment of the geologist. For this reason the roughness values are divided into three subjective classes: (1) smooth, (2) slightly rough, and (3) rough. The apertures are also only given as approximate values (e.g. < 0.01 m). In the text we have now revised or added the following sentences (lines 174 to 176): "*On the field scale, the discontinuity roughness was determined by subjective (qualitative) and visual evaluation of wavelength measurements with a tape line. The specified roughness classes correspond to the average roughness on the respective scale.*".

We have added the following sentence to describe the determination of the aperture (lines 176 to 177): "*The determined values correspond to the mean apertures determined by geometric (qualitative) analysis of the respective total discontinuity lengths.*".

**Line 217:** "Plane parallel"? Should this be "bedding parallel", or perhaps more accurately cut perpendicular to the core axis?

**Author's reply:** Thank you very much for the hint. Here is a word-mix of German and English. We have changed the sentence as follows (lines 222 to 223): *''All samples were saw-cut perpendicular to the core axis and their end faces were ground square to the maximal possible length l.''.*

**Line 276:** Are these mean or average orientations? If so, please specify that.

**Author's reply:** We have specified the expression and the corrected sentence reads (lines 280 to 282): "*In Wuppertal the sets were grouped into the arithmetic mean fracture azimuth directions of strike 172° (181 discontinuities), 55° (142 discontinuities), and 105° (138 discontinuities) (Fig. 4c,d; Table 6).*".

**Line 368:** "Highly related" is unclear. I would say they are "consistent with" folding.

**Author's reply:** Thank you very much for pointing out these linguistic details. We have adjusted the sentence.

**Lines 383 to 384:** First you say this is basically unknowable, but then you speculate on a cause. Why not just state more simply that the exact cause is unknown, but it may be due to local stress anomalies accumulated through time?

**Author's reply:** Thank you. We have adjusted the following sentences accordingly (lines 389 to 392): "The cause of the slightly different fault strike directions in Hönnetal cannot be specified according to the current state of scientific knowledge and is therefore unknown. However, the fracture formation could be explained by various local and temporal stress anomalies and different formation times.".

**Line 402:** "Towards THE core of THE reef formation"??

**Author's reply:** Thank you. We have corrected the sentence.

**Lines 461 to 462:** I don't get this comparison. Fractures that are filled with cement are by definition not open.

**Author's reply:** You' re right. We meant partially filled fractures, but unfortunately that was not in the text. We have now added this (lines 470 to 471): "*This is particularly true for fractures that are partially filled, for example, with cement (Laubach et al., 2004).*".

**Line 465:** "Before utilizing" what, exactly?

**Author's reply:** True, at this point the reference is unclear. We have made the sentence more precise (lines 474 to 476): "*Before utilization of geothermal energy from the potential deep fractured carbonates in the Rhine-Ruhr Area, the material changes between host rock and vein material might predict fracture orientations in depth by further geophysical driven studies (e.g., density changes or reflection coefficients).*".

**Lines 472 to 473:** Does a consideration of fracture connectivity not by definition require a consideration of dispensation in orientation? For example, fractures that have strongly consistent orientations are likely to exhibit poor connectivity.

**Author's reply:** You are right that the noted sentence is not appropriate. In this sentence we have now added that it is of course about the comparison of different fracture sets, which are distinguished by definition by different orientation. The adjusted sentence reads (line 482 to 484): "*Comparing the discontinuity trace lengths with the true discontinuity spacing of the different fracture sets gives a first idea of the connectivity of the fracture network.*".

**Line 505:** Please change "conclude on" to "infer".

**Author's reply:** We have corrected the sentence.

**Line 517:** Please restate the spatial scale of the wavelength so that the reader may consider the localization of fracture porosity in the rocks.

**Author's reply:** Good comment. Based on the smallest measured S-wave velocity and the largest measured P-wave velocity we determined the smallest and largest wavelength for the investigated ultrasound frequency (1 MHz), respectively. We have inserted this range into the text (lines 527 to 528): "*On the scale of wavelength (1 to 7 mm), the samples are thus probably heterogeneous.*".

**Lines 519 to 521:** I do not follow this argument as written. Please clarify. How would locally distributed, heterogeneous fracture networks contribute to fluid flow? As written it sounds like fracture porosity at this scale is NOT connected?

**Author's reply:** We have rewritten the sentence to emphasize our statement more clearly (lines 528 to 531): "*However, under the assumption of constant porosity, a possibly existing heterogeneous and local distribution of the pore and crack volume can favor the interconnection of these volumes to large pores or opened cracks more than a homogeneous distribution of many small pores could. In this case heterogeneity possibly favors the fluid flow.*".

**Line 530:** I would rearrange to say its dipping at XX degrees and likely occurs at a depth of YYYY meters.

**Author's reply:** Thanks. We have adjusted the sentence (lines 538 to 540): "*The target horizon of interest is an approximately 150 m thick and widely distributed compacted limestone layer (Massenkalk) from Late Devonian and is dipping northwards at 30 to 40° and likely occurs at a depth of 4000 to 6000 m.*".

**Lines 534 to 535:** Suggest rephrasing to "Outcrop investigation suggests that this unit could act as a naturally fractured reservoir at depth", or similar.

**Author's reply:** Thank you for improving the language of our manuscript. We have rephased the sentence in lines 544 to 545: *"However, outcrop investigation suggests that these unitscould act as a naturally fractured reservoir at depth.''*.

**Lines 536 to 537:** Why would numerous tectonic events and fracture reactivation necessarily produce smooth fractures?

**Author's reply:** This statement is based on many studies that consider the wear of rough fractures as a function of repeated reactivation or shearing. As examples we would like to cite the work of Hudson & Dowding (Int. J. Rock. Mech. Min., 1990) and Belem et al. (Rock. Mech. Rock. Eng., 2008). We have not included the references in the conclusions, but we have specified in the sentence that frequent reactivation causes the fracture surfaces to wear and the roughness to decrease (lines 546 to 547): "*Considering the numerous tectonic events in the Rhine-Ruhr region and the resulting repeated reactivation of pre-existing fractures and thus surface wear and/or degradation, the occurrence of slightly smooth fracture surfaces in the reservoir is more likely.*".

**Line 538:** Suggest rephrase to "We suggest that GEOTHERMAL EXPLORATION EFFORTS focus on ..." or similar.

**Author's reply:** We have changed the sentence according to your suggestion (lines 548 to 550): *"We suggest that geothermal exploration efforts focus on fractures that are approximately NNW–SSE oriented, bearing in mind that this direction is approximately parallel to the main horizontal stress according to the World Stress Map.''*.

**Line 540:** This sentence clashes with the one before it. First you argue they are likely open, not you say many are likely closed. Did examination of the calcite veins at the surface indicate a history of periodic reopening? This should be easily detectable from the morphology of the calcite crystals (i.e. crack seal veins?).

**Author's reply:** Thank you for pointing out that we have obviously contradicted ourselves at this point. We have removed this sentence with a view to clarity. We have also added our observations in the field regarding crack seal in lines 287 to 289: *''Besides, several generations of repeated cracking and sealing were observed, that were located within the recorded veins in Hagen-Hohenlimburg. A precise analysis of the individual vein generations is currently the subject of recent studies and could not be provided within the scope of this study.''*.

**Line 546:** "Despite THIS".

**Author's reply:** Thank you. We have corrected the sentence.

[revised manuscript text omitted]